# Expansion of lysosomal capacity in early adult neurons driven by TFEB/HLH-30 protects dendrite maintenance during aging in *Caenorhabditis elegans*

**Ruiling Zhong[1,2], Claire E. Richardson** [ID][2]*

**1** Department of Integrative Biology, University of Wisconsin—Madison, Madison, Wisconsin, United States of America, **2** Department of Genetics, University of Wisconsin—Madison, Madison, Wisconsin, United States of America

* claire.richardson@wisc.edu

## Abstract

Lysosomes are essential for neuronal homeostasis, providing degradation and recycling functions necessary to support neurons' complex operations and long lifespans. However, the regulation of lysosomal degradative capacity in healthy neurons is poorly understood. Here, we investigate the role of HLH-30, the sole *Caenorhabditis elegans* homolog of Transcription Factor EB (TFEB), a master regulator of lysosome biogenesis and autophagy that is thought to predominantly function in the context of starvation or stress. We demonstrate that HLH-30 is dispensable for neuronal development but acts cell-intrinsically to expand lysosomal degradative capacity during early adulthood. Loss of HLH-30 leads to lysosomal dysfunction and delayed turnover of synaptic vesicle proteins from the synapse. Notably, we show that basal HLH-30 activity is sufficient to expand neuronal lysosomal capacity without nuclear enrichment, in contrast to the nuclear translocation associated with starvation- and stress-induced activation of TFEB and HLH-30. Furthermore, we show that neuronal lysosomal function declines with age in wild-type animals, and this corresponds to a decrease in basal HLH-30-mediated transcription. We further demonstrate that basal HLH-30 activity is crucial for neuron maintenance: lysosomal dysfunction due to inadequate HLH-30 activity leads to dendrite degeneration and aberrant outgrowths. In summary, our study establishes a critical role for HLH-30/TFEB in promoting lysosomal capacity to preserve neuronal homeostasis and structural integrity of mature neurons *in vivo*.

## Introduction

To sustain their long lifespans, complex morphologies, and diverse functions, neurons require robust mechanisms of cellular homeostasis. Lysosomes are key

**Data availability statement:** All microscopy files are available from the BioImage Archive (accession number S-BIAD2252).

**Funding:** This work was supported by the United States Department of Agriculture (https://www.usda.gov/) grant WIS05016 and National Institutes of Health (https://www.nih.gov/) grant R35-GM154869 to C.E.R. The funders had no role in study design, data collection and analysis, decision to publish, or preparation of the manuscript.

**Competing interests:** The authors have declared that no competing interests exist.

**Abbreviations:** Aβ, beta-amyloid; AID, Auxin-inducible degradation; ANOVA, analysis of variance; ARGO, analysis of red and green offset; ART, aligned rank transform; BFP, blue fluorescent protein; Cas, CRISPR-associated protein; CGC, Caenorhabditis Genetics Center; CPR, cystine protease related; Cre, cyclic recombinase; CRISPR, clustered regularly interspaced short palindromic repeats; CUP, coelomocyte uptake-defective; DA, dorsal A type motor neuron; FLP, flippase; FRT, Flp recombinase recognition target; GFP, green fluorescent protein; GTP, guanosine triphosphate; HLH, helix loop helix; K-NAA, 1-naphthaleneacetic acid potassium salt; LMP, lysosome-associated membrane protein; LoxP, locus of X-over P1; LUT, look up table; MiTF, microphthalmia transcription factor; mTORC1, mammalian target of rapamycin complex 1; NCBI, National Center for Biotechnology Information; NGM, nematode growth medium; NLS, nuclear localization signal; NUC, abnormal nuclease; PEST, proline, glutamic acid, serine, and threonine; PVD, posterior ventral process D; RFP, red fluorescent protein; SNG, synaptogyrin; SNT, synaptotagmin; SYX, syntaxin; Tau, tubulin-associated unit; TIR, transport inhibitor response; TFEB, transcription factor EB; VHA, vacuolar H ATPase; WT, wild-type.

effectors of cellular homeostasis. Using the dozens of enzymes within their acidic lumens, lysosomes degrade transmembrane proteins, protein aggregates, and organelles, promoting their proper abundance and quality control. In neurons, degradative lysosomes reside predominantly in the cell body [1–3]. Their cargoes are earmarked for degradation throughout the neuron primarily via the endolysosomal pathway, which sorts transmembrane proteins for degradation into late endosomes, or through autophagy, in which autophagosomes engulf organelles and protein aggregates [2,4–8]. Once transported to the cell body, these compartments fuse with lysosomes to generate endolysosomes and autolysosomes, respectively, where the cargo is degraded before the lysosomes reform, ready for the next load [4,9,10]. For neurons to maintain effective lysosomal degradation, their lysosomal capacity—the number and functionality of lysosomes—must be sufficient to meet their high demands for cellular turnover. Yet, despite its importance, how neuronal lysosomal capacity is regulated remains largely unknown.

A potential regulator for neuronal lysosomal capacity is Transcription Factor EB (TFEB), which regulates the expression of genes involved in autophagy and lysosome biogenesis [11–14]. The activity of TFEB is best understood in the context of nutrient sensing: in nutrient-rich conditions, the mammalian target of rapamycin complex 1 (mTORC1) phosphorylates TFEB, leading to its cytosolic retention and inactivation. During starvation, mTORC1 activity is inhibited, allowing dephosphorylated TFEB to enter the nucleus to induce the expression of endosomal, lysosomal, and autophagosomal genes [15–17]. Similarly, HLH-30, the sole *Caenorhabditis elegans* homolog of TFEB, regulates lysosome biogenesis and autophagy genes in response to stress [18,19]. Like mammalian TFEB, HLH-30 predominantly localizes to the cytoplasm under well-fed conditions [11,13,19]. However, starvation or various stressors promote its nuclear translocation in hypodermal and intestinal cells [20–22]. The role of TFEB/HLH-30 in neurons under non-stressed conditions remains largely unexplored.

Understanding the regulation of neuronal lysosomal capacity has important health implications, as declining endolysosomal and autophagic function is closely linked to neurodegenerative diseases [23–28]. Lysosomal dysfunction plays a central role in neurodegenerative diseases, where protein aggregates, which are normally targeted for lysosomal degradation, accumulate and cause cellular toxicity [29–31]. There is evidence that lysosomal dysfunction is also a pathology of normal aging, leading to speculation that age-associated lysosomal dysfunction in neurons may contribute to neurodegenerative disease. For example, lipofuscin—an autofluorescent mix of oxidized macromolecules—accumulates within neuronal lysosomes during aging, which is thought to be indicative of and/or cause impaired lysosomal function [32,33]. Aging has also been associated with the accumulation of late endosomes in the presynapses in both mouse and *Drosophila* neuromuscular junctions, which could arise from an impairment in lysosomal function [34,35]. A reduced fusion rate between autophagosomes and lysosomes, along with the failure to deliver autophagosome substrates into lysosomes in aged mice, indicates that neuronal autophagy declines with aging [36,37]. Furthermore, axonal transport of autophagosomes and late

endosomes, which relies on acidification from fusing with lysosomes, show defects in aging mice [38–42]. This evidence highlights the need for a better understanding of whether and how neuronal lysosomal capacity becomes inadequate in healthy aging, to inform interventions against neurodegenerative disease.

Here, we present direct evidence that neuronal lysosomal function declines during healthy aging in vivo. Next, we showed that HLH-30 functions neuron-intrinsically to promote adequate lysosomal capacity in adulthood. Loss of HLH-30 accelerates the age-related decline in lysosomal function, and it causes delayed degradation of typical cargoes for endo-lysosomal degradation—synaptic vesicle proteins—leading to a backlog of degradative materials at the synapse. Inter-estingly, we found that basal HLH-30 activity supports neuronal lysosomal capacity without enriched nuclear localization, in contrast to the nuclear enrichment associated with TFEB and HLH-30 activity during starvation and stress. Additionally, HLH-30-regulated gene expression shows a declining trend with age, suggesting that reduced HLH-30 activity contributes to age-associated lysosomal dysfunction. Finally, we demonstrated that neuronal HLH-30 is critical for neuronal mainte-nance, as its loss accelerates dendrite age-related degeneration and aberrant dendritic sprouting. Based on these results, we propose that TFEB/HLH-30 performs a crucial role in promoting lysosomal capacity to preserve neuronal homeostasis during adulthood *in vivo*.

## Results

### Evidence for declining lysosome function during aging in *C. elegans* neurons

To test the hypothesis that neuronal lysosomal dysfunction is a pathology of aging in vivo, we examined the poste-rior ventral process D (PVD) neurons. These two bilaterally symmetric, glutamatergic sensory neurons provide an accessible model to interrogate lysosomal function with high resolution. Lysosomal degradation occurs predomi-nantly in the neuron cell body, and as in other organisms, the lysosomal compartments within *C. elegans* neurons mostly localize to the cell body [43]. We therefore focused on the PVD neuron cell body and assessed two estab-lished features of lysosomal dysfunction: increased luminal pH and buildup of degradative cargo within lysosomal compartments [44].

First, we developed a tool to measure relative lysosomal acidity. Healthy lysosomes have a luminal pH of about 5, and a higher pH indicates lysosomal dysfunction [3,45]. NUC-1, the ortholog of human DNAse II, is a luminal lysosome resident and a previously validated marker for lysosomal compartments—lysosomes, endolysosomes, and autolyso-somes—in *C. elegans* [46]. We co-expressed NUC-1::GFP and NUC-1::RFP alongside a BFP morphology marker in PVD neurons (Figs 1A, S1A, and S2A). GFP fluorescence is partially quenched in the acidic environment of lysosomal com-partments, while RFP intensity is unaffected by the lysosomal pH and is used for normalization. Therefore, the GFP/RFP ratio represents relative acidity: a higher ratio indicates reduced acidity (i.e., higher pH). We first validated our tool using wild-type (WT) worms continuously treated with 0.03 μM concanamycin A, which inhibits vacuolar H$^+$ ATPase, from Day 0 of adulthood. Relative lysosomal acidity is decreased in Day 6 adults post-treatment compared to untreated animals, demonstrating the serviceability of our tool (Fig 1B). Lysosomal acidity in Day 3 concanamycin A-treated adults is not significantly different from controls, indicating that this low dose of concanamycin A requires time to get into neurons and impede lysosomal function. We also injected Day 3 adult WT worms with 25 μM concanamycin A, and we found that the relative lysosomal acidity is decreased 3 h post-concanamycin A injection compared to mock drug injection. This indicates that a high, acute dose of concanamycin A is able to disrupt lysosomal acidity in early adulthood, as expected, further validating the tool (Fig 1C). We then examined WT worms at different ages and found that relative neuron lysosome acid-ity decreases in aged worms by Day 9 of adulthood (Fig 1D). For comparison, the median life span of worms is about 2 weeks (S2B Fig) [19,47–49].

Next, we aimed to assess the efficiency of lysosomal degradation by quantifying the accumulation of a degradative cargo, the synaptic vesicle protein Synaptotagmin/SNT-1, within lysosomal compartments in the neuron cell body. We measured the fluorescence intensity of endogenously tagged SNT-1::GFP that co-localized with NUC-1::RFP, which

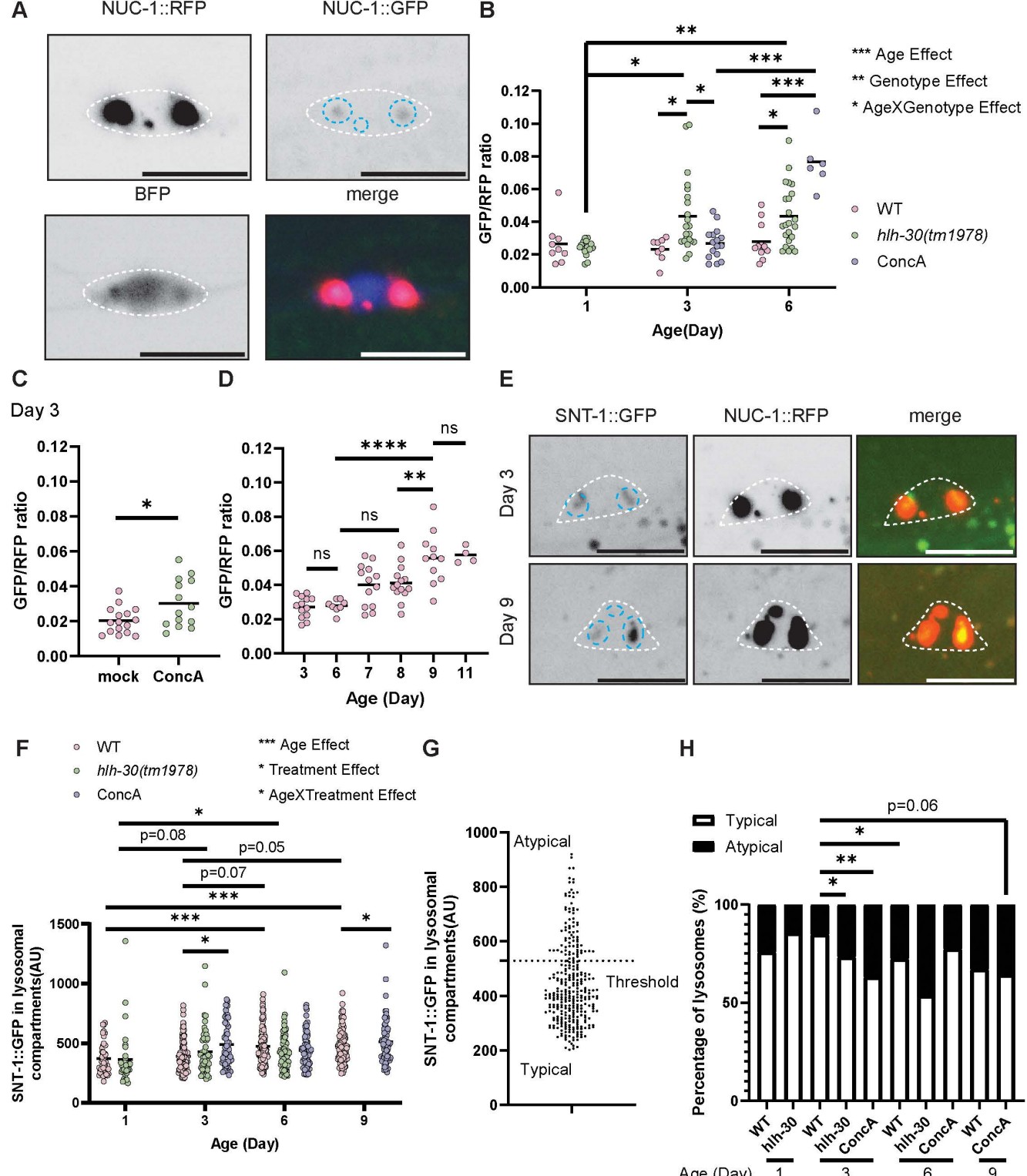

**Fig 1. PVD neurons show reduced lysosome function during aging and in the *hlh-30* mutant. (A–D)** A fluorescent reporter for luminal acidity of neuronal lysosomal compartments shows that acidity decreases with aging and in the *hlh-30* mutant. **(A)** Example images of the PVD neuron cell body in a Day 3 adult carrying the *carEx6* transgene, which stably expresses all three constructs shown. White dashed lines indicate the edge of the cell body,

blue dashed lines indicate the edge of the lysosomal compartments. Scale bar = 10 μm. **(B)** Quantification of NUC-1::GFP/RFP ratio using the *carEx6* transgene. \*\*\**P* < 0.001, \*\**P* < 0.01, \**P* < 0.05, pairwise comparisons not labeled show no significant difference, Two-way ANOVA with Sidak post-test. **(C)** The NUC-1::GFP/RFP ratio is increased 3 h after injecting concanamycin A into early-adult (Day 3) worms compared with mock drug injection. \**P* < 0.05, two-tailed Student *t* test. **(D)** The NUC-1::GFP/RFP ratio increases with age in WT. \*\*\*\**P* < 0.0001, \*\**P* < 0.01, ns: not significant, One-way ANOVA with Tukey post-test. **(E–H)** Images and quantification of a representative degradative cargo for lysosomes, the synaptic vesicle transmembrane protein SNT-1, within lysosomal compartments. **(E)** Representative images of WT animals. White dashed lines indicate the edge of the cell body, blue dashed lines indicate the edge of the lysosomal compartments. Scale bar = 10 μm. **(F)** SNT-1::GFP shows increased accumulation in lysosomal compartments with age. \*\*\**P* < 0.001, \**P* < 0.05, linear mixed-effects model with animal as a random effect and genotype/treatment and age as fixed effects, followed by pairwise comparisons with Tukey correction. Statistical comparisons were performed on log-transformed data to meet model due to heteroscedasticity in the untransformed data. **(G)** The amount of degradative cargo SNT-1 in individual lysosomal compartments shows a bimodal distribution. Threshold = 529 AU. **(H)** Aging, chronic concanamycin A exposure, and the *hlh-30* mutant each result in an increased portion of lysosomal compartments with an atypically high amount of SNT-1::GFP. \*\* *P* < 0.01, \* *P* < 0.05, pairwise comparisons not labeled show no significant difference, Fisher's exact test with Holm correction. ConcA, concanamycin A. The data underlying the graphs shown in the figure can be found in S1 Data.

represents SNT-1 protein that has been sorted into a lysosomal compartment but has not been fully degraded, and the GFP has not been fully quenched (Figs 1E–1H and S1B) [50,51]. The intensity of SNT-1::GFP in lysosomal compartments increases with age, which could be indicative of age-related lysosomal dysfunction (Fig 1F). Chronic exposure to concanamycin A does not exacerbate this accumulation (Fig 1F). This suggests that the age-associated increase of SNT-1::GFP within lysosomal compartments may not, in itself, be a pathology caused by lysosome dysfunction. On the other hand, autophagosome maturation and retrograde transport of lysosomal degradative cargoes requires acidification by the vacuolar $H^+$ ATPase, meaning that concanamycin A may not only reduce lysosome acidity but also disrupt delivery of retrograde cargoes, including SNT-1 to the lysosome [52].

By plotting the SNT-1::GFP fluorescence intensity of all lysosomal compartments from WT, we observed a bimodal distribution, suggesting the presence of two distinct lysosomal compartment populations: typical lysosomal compartments and atypical lysosomal compartments (Fig 1G). The percentage of atypical lysosomal compartments increases by Day 6 of adulthood (Fig 1H). Chronic concanamycin A treatment caused an increase in the percentage of atypical lysosomal compartments by Day 3 of adulthood, to a degree similar to that of older, untreated WT animals (Fig 1H). Considering that both aging and concanamycin A treatment led to an increased percentage of atypical lysosomal compartments, we speculate that this phenotype arises from reduced degradative functionality of the lysosomes. An alternative interpretation is that the same phenotype arises with concanamycin A treatment and older WT animals via two distinct mechanisms. In this alternative model, the increased percentage in older animals could arise from an increase in the amount of degradative cargo delivered to the lysosomes. These two models—age-associated lysosome dysfunction versus increasing degradative flux—are considered further below.

## HLH-30 is required for adult neuronal lysosomal function

We next aimed to understand how neurons generate adequate lysosomal function in early adulthood. TFEB is a master regulator of lysosome and autophagosome biogenesis in mammalian cells, but it has been predominantly shown to function during starvation and stress [53]. Whether and how TFEB functions in healthy neurons *in vivo* is unknown. We first asked whether loss of *hlh-30*, the *C. elegans* homolog of *TFEB*, affects the late endosome, endolysosome, and lysosome population in the neuron. The *hlh-30(tm1978)* allele is a frameshifting deletion and putative null allele. The *hlh-30(tm1978)* mutant, hereafter referred to as the *hlh-30* mutant, is viable and displays WT animal morphology and behavior [19]. The morphology of the PVD neuron in the *hlh-30* mutant appears indistinguishable from WT at Day 1 of adulthood (Fig 4A–4C). The RAB-7 GTPase is a trafficking factor of late endosomes and autophagosomes, and it localizes to late endosomes, endolysosomes, and autolysosomes [38]. Using PVD-expressed mCherry::RAB-7, we found that the number of RAB-7-labeled compartments in the neuron cell body increases between Day 0 of adulthood (the L4 larval stage) to Day 3 of adulthood (Fig 2A and 2B) [54]. In the *hlh-30* mutant, the number of RAB-7 labeled compartments is similar at Day 0 but significantly reduced at Day 3 of adulthood compared to WT (Fig 2B). This reduction at Day 3 can be

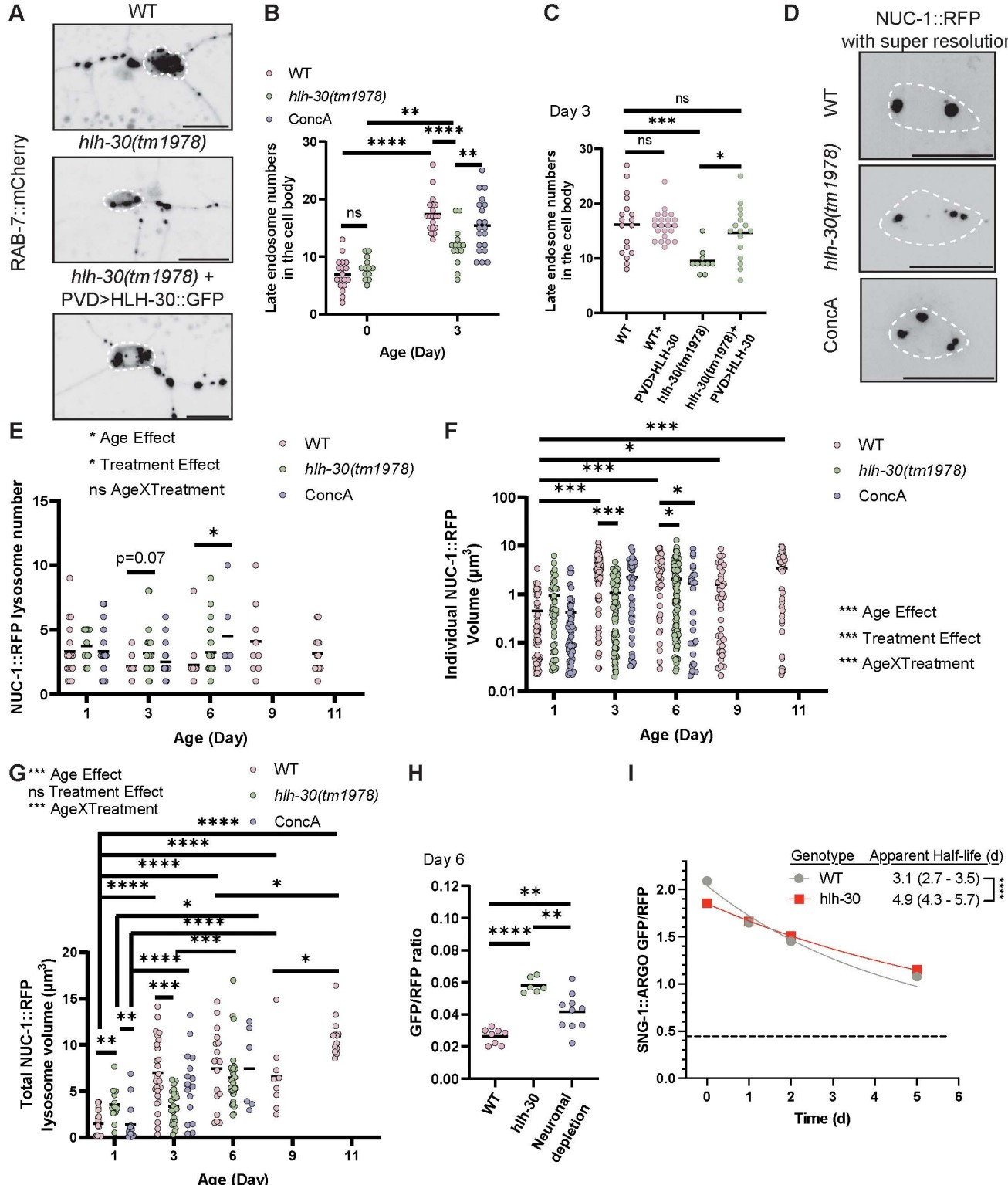

**Fig 2. HLH-30 functions cell-intrinsically to promote neuronal lysosomal function in early adulthood. (A–C)** The *hlh-30* mutant has fewer RAB-7::mCherry-labeled endosomes in the PVD neuron cell body by Day 3 of adulthood compared to WT, and this reduction can be rescued by PVD-specific expression of HLH-30. White dashed lines indicate the edge of the cell body. Scale bar = 10 μm. ****$P < 0.0001$, ***$P < 0.001$, **$P < 0.01$,

*$P < 0.05$, ns: not significant, One-way ANOVA with Tukey post-test. (D–G) Quantification of cell body lysosomal compartment number and size using super-resolution fluorescence microscopy of NUC-1::RFP. (D) Example images of lysosomal compartments in PVD neuron cell body in WT, *hlh-30* mutant, and WT with chronic concanamycin A treatment at adult Day 3. White dashed lines indicate the edge of the cell body. Scale bar = 10 μm. (E) Number of lysosomal compartments varies between individual neurons within the same age/genotype/treatment group and does not show strong changes by age, genotype, or treatment. *$P < 0.05$, pairwise comparisons not labeled show no significant difference, Two-way ANOVA with Tukey post-test. (F) The size of the largest lysosomal compartments increases in early adulthood, between Day 1 and 3, then remains stable in WT animals. Both the *hlh-30* mutant and concanamycin A treatment cause reduced size of the largest lysosomal compartments by mid-adulthood. ***$P < 0.001$, *$P < 0.05$, pairwise comparisons not labeled show no significant difference, linear mixed-effects model with animal as a random effect and genotype/treatment and age as fixed effects, followed by pairwise comparisons with Tukey correction. Statistical comparisons were performed on log-transformed data to meet model assumptions due to heteroscedasticity in the untransformed data. (G) Total lysosomal compartment volume in the neuron cell body increases between adult Day 1 and Day 3 in WT animals but not in the *hlh-30* mutant. ****$P < 0.0001$, ***$P < 0.001$, **$P < 0.01$, *$P < 0.05$, pairwise comparisons not labeled show no significant difference. Statistical comparisons were performed using a linear regression model on square root-transformed data to meet model assumptions, followed by pairwise comparisons with Tukey correction. (H) Neuronal depletion of HLH-30 shows increased lysosomal compartment luminal pH compared to WT at adult Day 6. ****$P < 0.0001$, **$P < 0.01$, One-way ANOVA with Tukey post-test. (I) SNG-1::ARGO turnover rate at the synapses, calculated from one-phase exponential decay curves. ****$P < 0.0001$, Extra sum-of-squares $F$ test. ConcA, concanamycin A. The data underlying the graphs shown in the figure can be found in S1 Data.

rescued by expressing *hlh-30::gfp* specifically in the PVD neurons, indicating that HLH-30 regulates the lysosomal compartment abundance cell-autonomously (Fig 2C). We observed the same trends when we quantified the abundance of endogenously-tagged GFP::RAB-7 compartments in WT versus *hlh-30* mutant animals, indicating that the accumulation of RAB-7-labeled lysosomal compartments in adult worms and their reduced abundance in *hlh-30* mutants are not due to an overexpression artifact (S2C Fig ) [54]. These data suggest that HLH-30 promotes late endosome, endolysosome, and/or autophagosome abundance in early adulthood but is dispensable in development.

Chronic exposure to concanamycin A did not result in significantly fewer mCherry::RAB-7-labeled endosomes in the cell body (Fig 2B). This may be because it takes more time for this low dose of concanamycin A to get through the cuticle and impact lysosomal compartments.

Next, we quantified NUC-1::RFP-labeled lysosomal compartment number and volume in WT versus *hlh-30* mutants (Figs 2D–2G and S2D). We used super-resolution microscopy for the improved ability to detect clusters of closely spaced lysosomal compartments versus large compartments compared with standard confocal microscopy (S2E Fig).

In WT, we observed a wide range of lysosomal compartment numbers and sizes between neurons in age-matched animals (Fig 2E and 2F). Within the neuronal cell body, both small lysosomes and larger endolysosomes or autolysosomes can be observed at any given time (Figs 2D and S2E). The mean size of individual NUC-1::RFP-labeled lysosomal compartments increases between Day 1 and 3 of adulthood, then remains stable through Day 11 (Fig 2F). This difference is driven by the larger size of the largest lysosomal compartments in older adults. The total lysosomal compartment volume within the neuron cell body also increases between Day 1 and 3 of adulthood, by close to 6-fold (from 1.1 μm$^3$ to 6.4 μm$^3$, $P < 0.0001$) (Fig 2G). For comparison, the neuron cell body size is similar between Day 1 and Day 3 adults but increases significantly between Day 3 and Day 6 (S2F Fig).

We predicted that the number of NUC-1::RFP-labeled compartments would be decreased in the *hlh-30* adults, showing a similar trend as mCherry::RAB-7-labeled late endosomes, endolysosomes, and autophagosomes, but this was not the case (Fig 2E). The average individual lysosomal compartment size in the *hlh-30* mutant appeared similar to WT at adult Day 1, and the total lysosomal compartment volume within each neuron was, unexpectedly, significantly greater in the Day 1 *hlh-30* mutant (3.3 μm$^3$) compared to WT (1.1 μm$^3$, $P < 0.01$) (Fig 2F and 2G). Unlike in WT, however, the *hlh-30* mutant did not substantially increase either the size of individual lysosomal compartments or the total lysosomal compartment volume between Days 1 and 3 (Fig 2F and 2G). These results suggest that HLH-30 is required to expand lysosomal compartment capacity in the neuron cell body between Days 1 and 3 of adulthood.

We quantified the number and size of NUC-1::RFP-labeled lysosomal compartments in WT animals chronically treated with concanamycin A to assess how lysosomal compartment behavior changes when lysosomal acidity is disrupted. Similar to the *hlh-30* mutant, concanamycin A treatment led to a reduction in the average volume of individual

lysosomal compartments by Day 6 of adulthood (Fig 2F). At the same time point, concanamycin A-treated animals showed an increase in lysosomal compartment number, suggestive that concanamycin A may impair lysosome fusion dynamics (Fig 2E).

Based on these results, we hypothesized that HLH-30 is required to generate adequate lysosomal capacity in adult neurons. If this is the case, then *hlh-30* mutants might exhibit accelerated neuronal lysosomal dysfunction in adulthood compared to WT. Using the PVD lysosome acidity reporter (Fig 1A–1D), we found that the NUC-1::GFP/RFP ratio is higher in *hlh-30* mutants compared to WT at Day 3 and Day 6 of adulthood, indicating that lysosomal compartment acidity is indeed disrupted in *hlh-30* mutants (Fig 1B). To test whether the impact of HLH-30 on lysosomal acidity maintenance is cell-autonomous, we used the Auxin-inducible degradation (AID) system, wherein we inserted the *AID* sequence into the *hlh-30* genomic locus to tag the C-terminus of HLH-30 (S1F Fig). When the hormone auxin is added, the tissue-specifically expressed TIR1 E3 ubiquitin ligase recognizes the AID-tagged HLH-30 and promotes its degradation [55]. This technique allows us to deplete HLH-30 in specific tissues. Neuronal depletion of HLH-30 is able to deplete most of HLH-30 protein in PVD neurons (S3A–S3C Fig). Lysosomal acidity under neuronal HLH-30 depletion is significantly reduced compared to WT (Fig 2H). The most parsimonious interpretation of this result is that HLH-30 promotes neuronal lysosomal compartment acidity cell-autonomously, though it is possible that it functions non-cell-autonomously from other neurons (Fig 2H).

Next, we assessed accumulation of degradative cargo SNT-1::GFP in the lysosomal compartments of *hlh-30* mutant neurons (Fig 1F and 1H). We found that the percentage of lysosomal compartments with atypically high levels of SNT-1::GFP is increased in the *hlh-30* mutant by Day 3 of adulthood (Fig 1H). This is consistent with the model that lysosomal degradation is less efficient in the absence of HLH-30 function, though an alternate interpretation is that the amount of cargo that is delivered to lysosomal compartments for degradation is increased, in which case the efficiency of degradation could be unchanged.

To determine which of these models is more accurate, we set out to directly test the notion that the HLH-30-mediated expansion of lysosomal functionality is necessary for efficient turnover of degradative cargo. To do this, we applied our recently developed Analysis of Red-Green Offset (ARGO) method, wherein we endogenously tagged another characteristic cargo of lysosomal degradation, the synaptic vesicle protein Synaptogyrin/SNG-1, with both GFP and RFP (S1C and S4A Figs) [56]. The ARGO method involves a pulse-chase component in which the gene encoding *gfp* is excised via Cre/LoxP recombination with the pulse, and then the neuron is periodically imaged to quantify the ratio of GFP/RFP intensity at the synapses during the chase. A one-phase exponential decay curve was fitted to these data to calculate the presynaptic half-life of SNG-1. We induced the visualization of SNG-1 specifically in the cholinergic motor neuron Dorsal A type motor neuron 9 (DA9) using FLP/FRT recombination. We selected the DA9 neuron for this experiment because A) it allows us to investigate the turnover dynamics of a synaptic vesicle protein, and the DA9 neuron serves as a well-established model for examining presynapses; and B) we aimed to determine whether HLH-30 promotes lysosomal capacity broadly across neurons or specifically in PVD neurons.

First, we examined steady-state SNG-1::ARGO fluorescence intensity in control animals that were not "pulsed." We observed no difference in presynaptic localization, organization, or number between *hlh-30* mutants and WT animals (S4B and S4C Fig). The presynaptic fluorescence intensity of SNG-1::ARGO RFP is slightly decreased in the *hlh-30* mutant compared to WT, suggesting a slight reduction in synaptic vesicle numbers per synapse (S4B and S4D Fig). Consistent with the framework that endolysosomal protein degradation occurs in the neuron cell body, we have observed that endosomes in the cell body show a lower SNG-1::ARGO GFP/RFP fluorescence intensity ratio compared to the presynapses in WT animals (S4E and S4F Fig, comparison not showed) [56]. In these cell body-localized endosomes, the GFP fluorescence from SNG-1::ARGO is partially quenched, indicative of the low pH environment of degradative endolysosomes (S4E Fig) [56].

To test this interpretation, we injected Day 3 adults with concanamycin A, imaged them 3 h post-injection, and quantified SNG-1::ARGO GFP/RFP ratio at the DA9 neuron cell body and synapses. Indeed, concanamycin A injection increases

SNG-1::ARGO GFP/RFP ratio of endosomes in the cell body (S4G Fig). We also noticed that the GFP/RFP ratio in the cell body is lower than at the synapses in *hlh-30* mutants, suggesting that the location of SNG-1 degradation—lysosomal compartments in the neuron cell body—is not altered in the *hlh-30* mutant (S4E, S4F, S5A, and S5C Figs, comparison not shown). Furthermore, concanamycin A injection increases the GFP/RFP ratio at the synapses (S4H Fig). Previous research indicates that synaptic vesicle proteins are sorted for degradation into late endosomes at the presynapses [7,8,57]. Our result likewise indicates that SNG-1 is sorted for degradation into acidic compartments—likely late endosomes—at the presynapses, causing some GFP to quench.

Next, with the ARGO pulse at Day 2 of adulthood, we found that the SNG-1 presynaptic turnover rate is slower in the *hlh-30* mutant (half-life = 4.9 d, 95% C. I. 4.3–5.7 d) compared to WT (half-life = 3.1 d, 95% C. I. 2.7–3.5 d) (Figs 2I, S5B, S5C, S5F, and S5G). Interestingly, the steady-state SNG-1::ARGO GFP/RFP ratio (in the absence of ARGO activation for pulse-chase imaging) is lower at the presynapses in the *hlh-30* mutant compared to WT at Day 2 and 4 of adulthood (Figs 2I and S4F). This result suggests that in the *hlh-30* mutant, a larger proportion of presynaptic SNG-1 resides in late endosomes rather than in synaptic vesicles compared to WT. Considering this change in steady-state presynaptic SNG-1::ARGO GFP/RFP along with the slower SNG-1 turnover rate in the *hlh-30* mutant, it suggests that there is a backup in the clearance of sorted-for-degradation SNG-1 from the presynapses in the *hlh-30* mutant.

The slower turnover of SNG-1 in the *hlh-30* mutant compared to WT motivated us to take a closer look at the steady-state fluorescence of SNG-1::ARGO-labeled compartments in the cell body, which are lysosomal compartments based on their low pH (S5A, S5D, and S5E Fig). Notably, in the *hlh-30* mutants, there are fewer and smaller SNG-1::ARGO-labeled endolysosomes in the neuron cell body, suggesting reduced delivery of SNG-1 to lysosomal compartments (S5D and S5E Fig). Further, the *hlh-30* mutants show a higher SNG-1::ARGO intensity ratio in the neuron cell body compared to WT, suggestive of dysfunctional endolysosomal degradation (S4E Fig).

Taken together, these results indicate that HLH-30 is required for maintaining neuronal lysosomal function in early adulthood, whereas loss of *hlh-30* leads to inadequate lysosomal degradation.

## Basal HLH-30 activity in neurons operates without nuclear enrichment and declines in aging

During starvation and stress, TFEB/HLH-30 translocates from the cytoplasm to the nucleus [20–22,58]. The subcellular localization of TFEB has been used as a key indicator of its activity status, wherein nuclear enrichment means it is active and cytosolic enrichment means it is inactive. We therefore wondered whether neuronal HLH-30 shows enriched nuclear localization when it is functioning in well-fed, unstressed young adults to expand neuronal lysosomal capacity. We used split GFP to visualize the sub-cellular localization of endogenously tagged HLH-30::GFP11 in neurons (Figs 3A and S1D) [59]. Neuronal HLH-30::GFP11 shows enriched nuclear localization upon heat stress, consistent with findings in other cell types (Fig 3A and 3B) [20,21]. However, HLH-30::GFP11 avoids the nucleus in the steady-state of both early- (Day 3) and mid-adult (Day 6) PVD neurons (Fig 3A and 3B). HLH-30::GFP11 is not completely absent from the nucleus, though, so it can still be affecting some transcriptional regulation (Fig 3A and 3B). When the animals are starved for 1 day, neuronal HLH-30::GFP11 also avoids the nucleus (Fig 3A and 3B). We speculate that the PVD neurons might be protected from starvation, preventing them from experiencing or sensing it during our starvation manipulation. Recent research suggests that HLH-30 activity is not always correlated with its nuclear localization [60]. Our data suggest that, although acute stress can increase neuronal HLH-30 activation by promoting its nuclear localization, the basal HLH-30 activity that expands lysosomal capacity in healthy, unstressed adulthood occurs in the absence of nuclear enrichment.

We hypothesized that basal HLH-30 activity declines in aged neurons, contributing to the aging-associated reduced lysosomal function. To test this hypothesis, we generated a real-time fluorescent transcriptional reporter of *rab-7*, wherein we express *GFP11::nls::PEST* from the endogenous promoter-plus-enhancers of *rab-7* and use neuronally-expressed *GFP1-10* to reconstitute GFP fluorescence specifically in the neurons (S1E Fig). *RAB7* is a known transcriptional target of TFEB in mammalian cells [12]. HLH-30 shares a similar bHLH binding sequence as TFEB, and there are several predicted

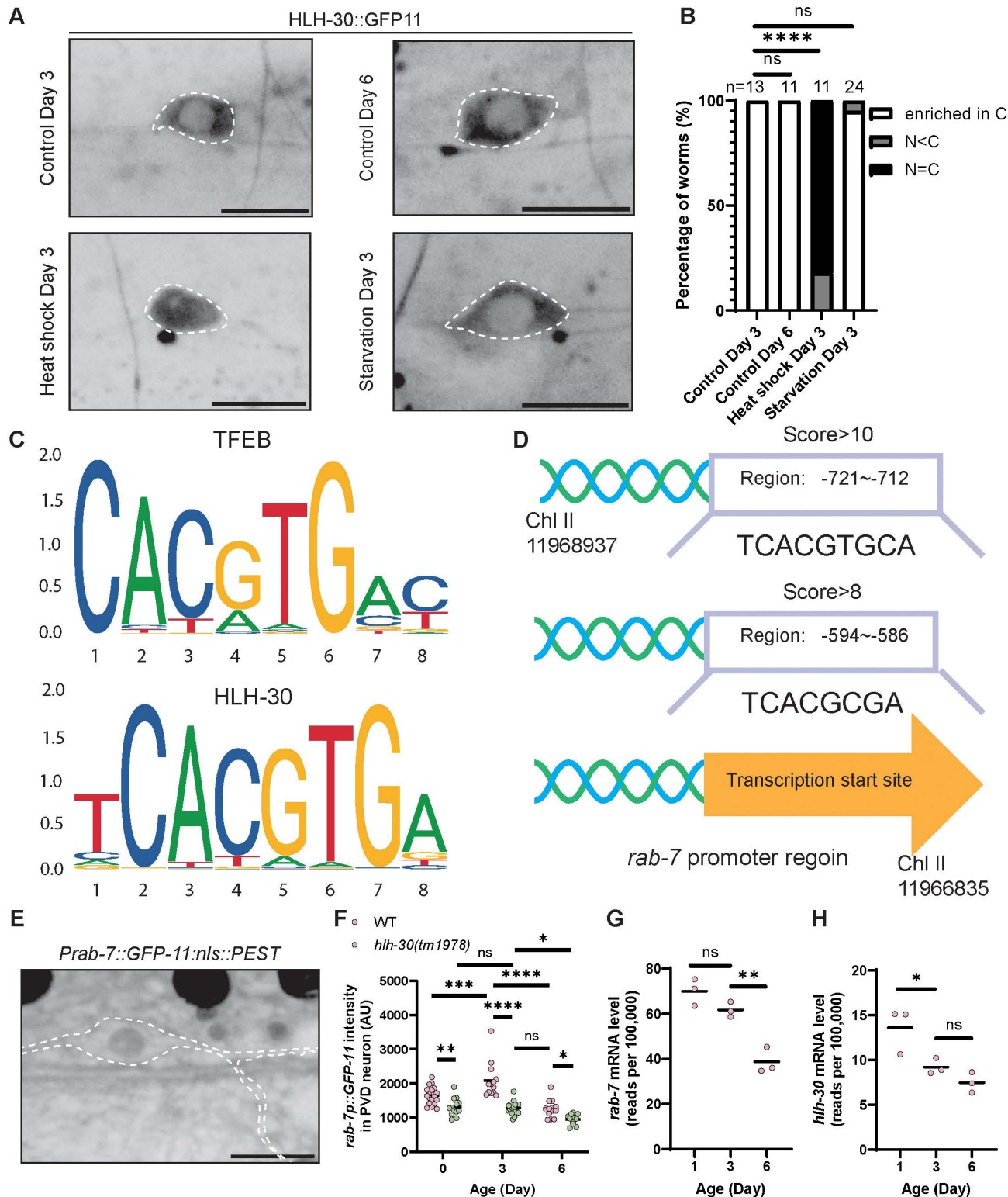

**Fig 3. Basal HLH-30 activity is not accompanied by enriched nuclear localization and declines during neuronal aging. (A, B)** Endogenous HLH-30, visualized with split GFP, is mainly localized in the neuron's cytoplasm in unstressed adulthood. Scale bar = 10 µm. ****$P < 0.0001$, ns: not significant, Fisher's exact test. **(C)** The bHLH binding sequence is similar between human TFEB and *Caenorhabditis elegans* HLH-30 [61]. **(D)** High score of

HLH-30 binding sequences is found in the *rab-7* promoter-plus-enhancer region. **(E, F)** A fluorescent reporter of *rab-7* transcription shows that PVD-specific *rab-7* transcription increases in early-adults (Day 3) and decreases during aging in WT. In the *hlh-30* mutant, the PVD-specific *rab-7* transcription level is lower compared to WT at the same age, and there is no difference between Days 0 and 3. There is still a slight decrease between Days 3 and 6. Scale bar = 10 µm. The outline of the PVD neuron was generated using the PVD > bfp fluorescence in this strain, which is not shown. ****$P < 0.0001$, ***$P < 0.001$, **$P < 0.01$, *$P < 0.05$, ns: not significant, Two-way ANOVA with Tukey post-test. **(G)** Relative *rab-7* mRNA abundance from whole-worm transcriptomic data shows reduced *rab-7* mRNA level in aging WT worms. **$P < 0.01$, ns: not significant, One-way ANOVA with Tukey post-test [62]. **(H)** Relative *hlh-30* mRNA abundance from whole-worm transcriptomic data shows reduced *hlh-30* mRNA level in aging WT worms. *$P < 0.05$, ns: not significant, One-way ANOVA with Tukey post-test [62]. The data underlying the graphs shown in the figure can be found in S1 Data.

binding sites of HLH-30 in the promoter and enhancer region of *rab-7* (Fig 3C and 3D) [61]. The *GFP11::nls::PEST* is transcribed together with *rab-7* into the same mRNA, which is then alternatively spliced to separate the *GFP11::nls::PEST* mRNA from the *rab-7* mRNA, to be translated separately. The PEST sequence targets protein for rapid degradation, so the fluorescence intensity of GFP11 reports the transcription level of *rab-7* within a short period, instead of the cumulative transcription level [63]. The NLS (nuclear localization signal) directs GFP11 to the nucleus, facilitating image quantification [64]. The transcription level of *rab-7* in WT PVD neurons increases in early adulthood (Day 3), corresponding with the expansion of the number of late endosome and lysosomal compartments. Then, it decreases in older neurons (Day 6) by 40 ± 5%, anticipating the age-associated decline of lysosomal function (Fig 3E and 3F). In *hlh-30* mutants, the transcription level of *rab-7* in the neuron is always lower compared to WT at the same age (Fig 3F). Furthermore, in contrast to WT animals, we detected no difference in *rab-7* transcription level between Day 0 and Day 3 in *hlh-30* mutant (Fig 3F). The fluorescence intensity of the reporter decreases between Day 3 and 6 in the *hlh-30* mutant by 26 ± 4%. This indicates that some of the decrease observed in WT is due to reduced HLH-30 activity, and there is also an *hlh-30*-independent mechanism that contributes to the declining *rab-7* expression between Day 3 and 6 (Fig 3F).

We analyzed pre-existing RNAseq data from whole animals to assess whether the changes we observed in neuronal *rab-7* transcription during adulthood extend to the whole animal [62]. The whole-animal *rab-7* mRNA level decreases from Day 3 to Day 6, consistent with the decrease observed with our neuron-specific transcriptional reporter (Fig 3G). On the other hand, there is no increase in the *rab-7* mRNA level in early adulthood, suggesting that the expansion of lysosomal capacity during early adulthood may be neuron-specific. We next examined the mRNA levels of five additional genes, *cpr-1*, *cpr-5*, *vha-5*, *lmp-2*, and *syx-17*, that are well-validated transcriptional targets of HLH-30 [65]. mRNA levels of *cpr-5*, *lmp-2*, and *syx-17* show a statistically significant decrease during aging. Therefore, 4 out of 6 HLH-30 transcriptional targets show reduced expression in aging, by bootstrapping analysis, only 3.5% of randomly selected groups of 6 genes have at least 4 that decline in aging (S6A–S6F Fig). Moreover, the level of *hlh-30* mRNA itself also decreases during aging (Fig 3H). HLH-30 is predicted to regulate its own expression based on ChIP-seq data [66], echoing the self-transcriptional regulation of mammalian TFEB [67]. Together, these results suggest that HLH-30 transcriptional activity decreases with age, likely contributing to the age-associated neuronal lysosomal dysfunction.

## Loss of HLH-30 accelerates age-related decline in dendrite morphology maintenance

In WT worms, the PVD neuron displays a highly branched and organized dendritic arbor. The structure is developed by Day 0, and the PVD neuron can fully maintain organization in middle-aged adult, Day 6 worms (Fig 4A). However, in Day 6 *hlh-30* mutants, we observed disorganized dendrite outgrowths, what we refer to as the "sprouting" phenotype, and dendrite degeneration, in approximately 75% and 50% of the population, respectively (Fig 4A, 4D, and 4E). We did not observe any difference in degeneration between WT and the *hlh-30* mutant at Day 1 or 3 (Fig 4B). The sprouting phenotype is noticeably increased in the *hlh-30* mutant compared to WT by Day 3 of adulthood (Fig 4C). Previous works have shown that the PVD dendrite exhibits progressive degeneration in WT as they age [68,69]. We likewise observed progressive dendrite degeneration with age in WT animals, and the *hlh-30* mutant at Day 6 exhibits a similar level of dendrite degeneration as WT at Day 11 (Fig 4D). We also observed that the sprouting phenotype increases with age in WT

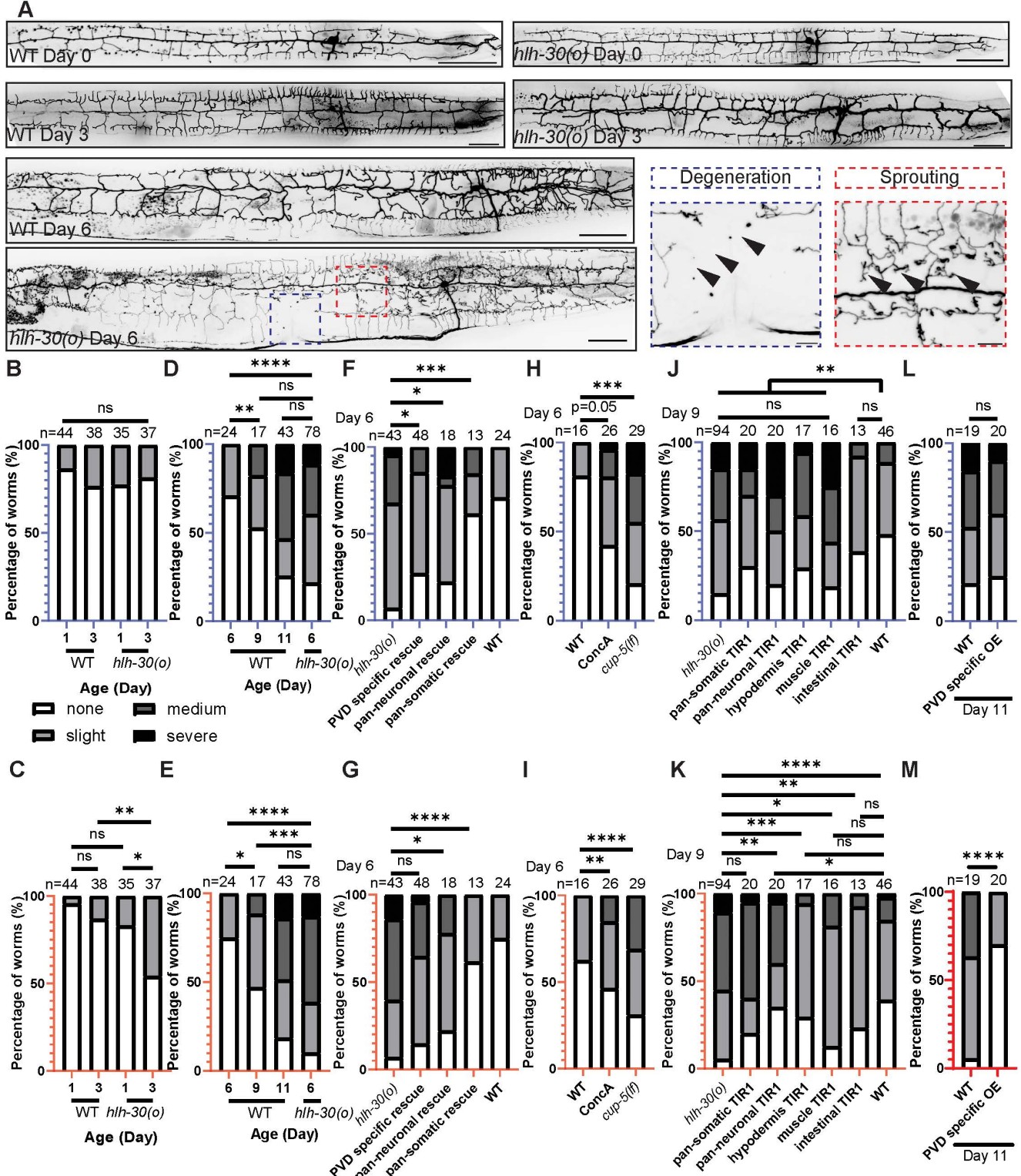

**Fig 4. Loss of HLH-30 accelerates age-related dendrite degeneration and sprouting. (A–E)** In WT adults, the PVD dendrite shows progressive dendrite degeneration (B and D) and sprouting (C and E) with age, and these phenotypes manifest earlier in the *hlh-30* mutant. Note that WT Day 11 shows similar penetrance and severity as the *hlh-30* mutant Day 6. In (A), the blue and red dashed boxes on the *hlh-30* Day 6 animal, which correspond

to the zoomed-in images on the right with the same outline colors, show examples of severe dendrite degeneration and severe sprouting, respectively. Scale bar = 50 μm (whole worm) or 10 μm (zoom in). ****$P < 0.0001$, ***$P < 0.001$, **$P < 0.01$, *$P < 0.05$, ns: not significant, Fisher's exact test. Arrowheads point to degeneration or sprouting (not all instances are indicated). **(F, G)** Pan-somatic expression of *hlh-30* rescues the dendrite degeneration (F) and sprouting (G) phenotypes in Day 6 *hlh-30* mutants, and PVD-specific or pan-neuronal *hlh-30* expression partially rescues the dendritic defects. ****$P < 0.0001$, ***$P < 0.001$, *$P < 0.05$, ns: not significant, Fisher's exact test. **(H, I)** WT animals exposed to concanamycin A treatment and *cup-5(lf)* mutant animals both show increased percentage and severity of dendrite degeneration (H) and sprouting (I) compared to WT at Day 6. ****$P < 0.0001$, ***$P < 0.001$, **$P < 0.01$, Fisher's exact test. **(J, K)** Depleting HLH-30 in the whole worm (except the germline) phenocopies the *hlh-30* mutant, showing increased dendrite degeneration (J) and sprouting (K) compared to WT at the same age. Note that for these experiments, all animals were cultivated in the presence of auxin, and they were scored at a later age than (A–I) because the auxin caused slowed growth and aging. **(J)** Pan-neuronal, muscle, and hypodermal-specific HLH-30 depletion all show increased degeneration compared to WT. **$P < 0.01$, ns: not significant, Fisher's exact test. **(K)** Pan-neuronal HLH-30 depletion causes increased sprouting compared to WT. ****$P < 0.0001$, ***$P < 0.001$, **$P < 0.01$, *$P < 0.05$, ns: not significant, Fisher's exact test. **(L, M)** PVD-specific overexpression of *hlh-30* in the WT genetic background does not have a significant impact on dendrite degeneration (L) and reduces severity of aging-associated sprouting (M). ****$P < 0.0001$, ns: not significant, Fisher's exact test. In all panels of this figure, the *hlh-30(tm1978)* mutant is labeled as *hlh-30(o)* for simplicity. The data underlying the graphs shown in the figure can be found in S1 Data.

animals, and the percentage of animals showing sprouting at Day 11 in WT is similar to that in the *hlh*-30 mutant at Day 6 (Fig 4E). These results suggest that loss of HLH-30 causes early-onset and accelerated neuronal aging in PVD neurons. Expression of wild-type *hlh-30* from a pan-somatic promoter was able to rescue the level of both sprouting and dendrite degeneration in the *hlh*-30 mutant to those observed in WT at Day 6, confirming that HLH-30 is necessary for maintaining dendrite morphology in aging (Fig 4F and 4G).

In addition to regulating lysosomal genes, both TFEB and HLH-30 regulate genes that are not involved in the autophagy or lysosomal pathway [65,70–72]. Given our results showing that HLH-30 promotes lysosomal functionality in *C. elegans* neurons (Figs 1, 2, S4, and S5), we hypothesized that both the dendrite sprouting and degeneration phenotypes result from decline of lysosomal function. To test this, we examined the PVD dendrite morphology in WT worms treated with V-ATPase inhibitor concanamycin A, which causes reduced acidity of lysosomal compartments (Fig 1B and 1C). Indeed, WT worms treated with concanamycin A show an increased percentage and severity of both sprouting and degeneration compared with controls (Fig 4H and 4I). CUP-5 is an ortholog of mammalian lysosomal channel protein MLN1/TRPML1; complete loss of CUP-5 is lethal, and partial loss of *cup-5* function results in reduced lysosomal functions [73,74]. We also assessed the PVD dendrite morphology in *cup-5(lf)* mutant animals (partial loss-of-function). Indeed, the *cup-5(lf)* mutant phenocopies the *hlh*-30 mutant by showing the sprouting and dendrite degeneration phenotypes at Day 6 (Fig 4H and 4I). These results indicate that both dendrite sprouting and degeneration are caused, at least in part, by decline of lysosomal function.

To determine the spatial requirement for HLH-30 to preserve dendrite morphology during aging, we used the Auxin-Inducible Degradation (AID) system, wherein we inserted the *AID* sequence into the *hlh-30* genomic locus to tag the C-terminus of HLH-30 (S1F Fig). When the hormone auxin is added, the tissue-specifically expressed TIR1 E3 ubiquitin ligase recognizes the AID-tagged HLH-30 and promotes its degradation [55]. This technique allows us to deplete HLH-30 in specific tissues (S3 Fig). Auxin delays worm growth, and the progression of age-related dendrite phenotypes irrespective of the presence of TIR1, so we performed all comparisons for this experiment in Day 9 adults grown on auxin (Fig 4J and 4K). We used pan-somatically expressed TIR1 to eliminate HLH-30 in the whole worm except the germline, which phenocopies the dendrite degeneration and sprouting phenotypes observed in the *hlh-30* mutant (Figs 4J, 4K, and S3A). Next, we depleted HLH-30 from four individual somatic tissues: neurons, hypodermis, muscle, and intestine (S3A–S3C Fig). Neuron, hypodermis and muscle-specific degradations all show a significant increase in dendrite degeneration compared to WT, and the severity is similar to that of the *hlh-30* mutant (Fig 4J). This suggests that HLH-30 is required in neurons, as well as in the two tissues that physically contact the PVD dendrite, to protect against dendrite degeneration. For the dendrite sprouting phenotype, pan-neuronal HLH-30 depletion shows an intermediate sprouting phenotype between *hlh-30* mutant and WT (Fig 4K). Muscle-specific HLH-30 degradation also results in a trend of higher percentage of the population exhibiting sprouting compared to WT, though it is not statistically significant (Fig 4K). These data indicate that HLH-30 is necessary in neurons and perhaps also in muscle to maintain organized PVD dendrites during aging.

Since HLH-30 functions cell-autonomously to expand lysosomal compartment abundance (Fig 2A–2C), we asked whether HLH-30 in the PVD neuron is likewise sufficient to protect against age-associated dendrite morphology defects. Indeed, overexpressing HLH-30 either specifically in PVD neurons or pan-neuronally can partially rescue the sprouting and dendrite degeneration phenotypes in the Day 6 *hlh-30* mutant (Fig 4F and 4G). These results are consistent with the results from the AID experiment (Fig 4J and 4K), and they indicate that HLH-30 is working in both cell-autonomous and non-cell-autonomous pathways to preserve adult dendrite morphology. The *hlh-30* mutant is reported to have a wild-type life span at 20 °C and a slightly shortened life span at 25 °C (S2B Fig) [19,75]. Importantly, the shortened life span at 25 °C cannot be rescued with neuron-specific HLH-30 expression [75]. Therefore, the rescue of the *hlh-30* mutant's dendrite degeneration and sprouting by PVD-specific or pan-neuronal *hlh-30* expression is not an indirect effect of life span differences between the strains.

Although PVD-specific overexpression of *hlh-30* in WT worms has no statistically significant impact on dendrite degeneration, it reduces the severity of sprouting at Day 11 (Fig 4L and 4M). These data indicate that HLH-30 is sufficient to partially protect against age-associated dendrite morphology defects, supporting a model in which the basal activity of HLH-30 during adult homeostasis is essential for preserving lysosomal function and thereby maintaining neuron integrity during aging (S7 Fig).

## Discussion

Here, we established a vital role for HLH-30/TFEB in maintaining lysosomal homeostasis in mature neurons in vivo. We proposed a model in which HLH-30/TFEB operates intrinsically within neurons of healthy young adults, in the absence of starvation or exogenous stress, to expand lysosomal functionality (S7 Fig). This likely aids in remediating the rising proteostasis demand during aging. In contrast to the stress conditions that promote HLH-30 translocation to the nucleus, HLH-30 functions in basal homeostasis while predominantly enriched in the cytoplasm. Our model further proposes that neuronal lysosomal degradative function declines in aged adults, in part due to declining HLH-30 activity, which in turn contributes to declining maintenance of dendrite morphology (S7 Fig).

Based on the rescue and tissue-specific AID experiments (Fig 4F, 4G, 4J, and 4K), it is likely that the dendrite degeneration and dendrite sprouting are regulated by HLH-30 through different mechanisms. Neuronal, muscular and hypodermal loss of HLH-30 all results in dendrite degeneration (Fig 4J). Together with the result that PVD specific or pan-neuronal rescue of HLH-30 partially rescue the dendrite degeneration phenotype in the *hlh-30* mutant (Fig 4F), we hypothesize that neuronal, muscular, and hypodermal HLH-30 all contribute to dendrite degeneration as addable effects (Fig 4). By contrast, of the individual tissue depletions, only neuronal depletion causes dendrite sprouting (Fig 4K). The PVD-specific expression of *hlh-30* does not significantly rescue the *hlh-30* mutant's dendrite sprouting defect (Fig 4G), yet PVD-specific overexpression of *hlh-30* in the WT background significantly reduces ectopic sprouting at Day 11 (Fig 4M). Taken together, these data point to both cell-autonomous and non-cell-autonomous roles for *hlh-30* in preserving dendrite morphology (Fig 4).

We demonstrated that neuronal lysosomal function declines with age in healthy, wild-type animals based on the failure to maintain the acidic lysosomal pH and the increased proportion of lysosomal compartments with atypically high degradative cargo accumulation (Fig 1) [44]. This decline in lysosomal function is consistent with previous findings across multiple species, including humans, mice, *D. melanogaster,* and *C. elegans,* showing that autophagic activity becomes impaired in aging neurons, potentiating the accumulation of neurodegenerative disease-associated protein aggregates and defective organelles [36,76–79]. Given that autophagic flux requires efficient lysosomal degradation, we propose that neuronal lysosomal dysfunction partially underlies this age-associated decline in autophagy. Importantly, comparing our finding to previous works, lysosomal dysfunction appears to precede other cellular pathologies of neuron aging, including synapse loss and axon morphology defects (Fig 1D, 1H) [80–82]. This sequence of events suggests that lysosomal dysfunction is an early molecular event in neuronal aging, similar to declining proteostasis, and that it likely contributes to impaired autophagic flux and declining proteostasis [83].

We propose that declining neuronal lysosomal function contributes to neuronal aging. Conversely, it also seems to be the case that neuronal aging contributes to declining neuronal lysosomal function. How aging causes inadequate HLH-30 activation, and what mechanisms function in parallel to HLH-30 to cause age-associated neuronal lysosomal dysfunction, would be interesting topics for future research.

TFEB is well-established as a master regulator of lysosome biogenesis, especially in response to starvation [11,13,14]. Its potential to enhance the autophagic clearance of pathogenic accumulations of lysosomal degradative cargo marks TFEB as a compelling therapeutic target for neurodegenerative disease. Indeed, overexpression of TFEB has demonstrated benefits in multiple experimental models of Alzheimer's and Parkinson's disease [84–89]. Our findings expand the framework for thinking about TFEB and neurodegenerative disease, suggesting that it is not only a potential therapeutic target but also a plausible underlying mechanism. Specifically, evolutionary conservation of the *C. elegans* function of HLH-30 would implicate TFEB in promoting adult neuron maintenance, and its declining activity in contributing to age-associated lysosomal dysfunction that may potentiate neurodegenerative disease [90]. Supporting the notion that this role in neuron homeostasis may be conserved, knockdown of TFEB/Mitf in *Drosophila* neurons results in an apparent blockage of autophagic flux in the adult fly brain in the absence of starvation or stress [91]. Further, brain-specific TFEB knockout in mice leads to increased Aβ and Tau protein accumulation, apoptotic cells, and axonal degeneration; it would be interesting to assess to what extent these phenotypes are due to loss of TFEB in neurons versus in glial cells [92,93].

It is generally accepted that when TFEB/HLH-30 is active, it shows a corresponding enriched nuclear localization [13,14,90,94]. However, our data indicate that HLH-30 promotes neuronal lysosomal capacity during steady-state maintenance while exhibiting enriched cytoplasmic localization (Figs 3A, 3B, 3F, 3G, and S6). Additionally, we observed reduced HLH-30 transcriptional output during aging without a corresponding change in its subcellular localization, further decoupling HLH-30 function from its cytosolic versus nuclear subcellular localization in the context of basal homeostasis. This result corroborates recent research suggesting that HLH-30 nuclear localization is not necessarily strongly correlated with its transcriptional output [60]. It would be interesting to investigate whether the transcriptional output of neuronal HLH-30 changes in composition as well as in strength during basal homeostasis versus stress [21]. Moreover, our results raise the consideration that a lack of nuclear enrichment for mammalian TFEB may not be a strong indicator that it does not have an important function in promoting a low level of basal transcription in a given context.

Our data indicate that neuron-intrinsic lysosomal dysfunction contributes to defective maintenance of dendrite morphology, causing both dendrite degeneration and sprouting. What mechanism links these phenotypes? For dendrite degeneration, one possibility is that lysosomal dysfunction causes a buildup of autophagosomes in the dendrite. These could locally disrupt transport or function of necessary cellular components, leading to degeneration. Indeed, autophagosomes are observed within the beading or swelling structures that are early signs of degeneration, and formation of autophagosomes within the neuron promotes dendrite degeneration [68]. A decline in autophagic flux might also lead to the accumulation of damaged mitochondria, increasing oxidative stress, or disrupting calcium homeostasis that could promote neurodegeneration [95]. Alternatively, or in addition, an accumulation of unwanted protein cargoes of endolysosomal degradation in the dendrite could disrupt structural integrity or ion homeostasis [96]. For dendrite sprouting, endocytic trafficking of dendrite guidance receptors is critical for maintaining proper receptor activity at the plasma membrane, so it is possible that disrupted lysosomal function leads to guidance receptor buildup on the plasma membrane, leading to the sprouting phenotype [97–99]. Interestingly, our results indicate that HLH-30 promotes dendrite maintenance both cell-intrinsically and non-cell-autonomously (Fig 4), so some extracellular pathways are also involved in both dendrite degeneration and sprouting. The observation that the *cup-5(lf)* mutant phenocopies the *hlh-30* mutant in regard to dendrite degeneration but exhibits less severe sprouting suggests that additional mechanisms beyond lysosomal defects may contribute to the sprouting phenotype (Fig 4H and 4I). TFEB and HLH-30 target genes involved in lysosomal biogenesis and autophagy are only a subset of their transcriptional targets; others include genes involved in metabolism, mitochondrial homeostasis, and protein transport [100,101]. Indeed, lysosome-independent functions of TFEB include regulating lipid metabolism during

starvation in the liver, glucose homeostasis during exercise in the muscle, inflammatory response in macrophages, and apoptosis in cancer cells [67,70,102,103]. Loss of one of the lysosome-independent function of HLH-30 may contribute to the dendrite sprouting phenotype.

Inadequate lysosomal degradation is expected to cause a buildup in degradative cargo within lysosomes [44]. Both aging and loss of HLH-30 lead to an increased proportion of lysosomal compartments with atypically high levels of lysosomal cargo Synaptotagmin/SNT-1 (Fig 1H). Further, we found that this buildup extends to the presynapse. Specifically, our data suggest that in the *hlh-30* mutant, the presynaptic half-life of SNG-1 is prolonged compared to WT, and this is accompanied by an increase in the proportion of presynaptic SNG-1 that resides in acidic late endosomes versus synaptic vesicles. This indicates that while SNG-1 is sorted into acidic late endosomes at the synapse in both WT and the *hlh-30* mutant, the clearance of these degradation-targeted SNG-1 molecules is delayed in the *hlh-30* mutant (Figs 2I, S4F, S5B, and S5C). Our findings are consistent with prior studies in *Drosophila* and mammalian neurons indicating that synaptic vesicle proteins are sorted for degradation at the presynapse, and they suggest that lysosome functionality contributes to regulating the half-life of presynaptic proteins [7,8,57,104]. Together, these findings suggest a mechanism wherein lysosome dysfunction causes impaired removal of presynaptic degradative cargoes, which could contribute to neuronal dysfunction.

Certain technical limitations of our study should be considered. Our measurements of lysosome size and number were based on a NUC-1::RFP transgene, which is overexpressed and may not fully reflect endogenous lysosomal dynamics. As such, we cannot rule out transgene effects on lysosomal numbers and morphology. Additionally, our HLH-30 neuronal rescue experiments used HLH-30::GFP overexpressed transgene, and the expression levels of HLH-30 in this system are unknown. Therefore, the potential influence of overexpression on the observed rescue effects cannot be excluded. Future studies using endogenously tagged reporters or single copy insertions will help validate these findings under physiological expression levels.

In summary, understanding how neurons adapt to proteostasis demands throughout aging is crucial for uncovering the mechanisms that support neuronal maintenance. Our work identifies HLH-30 as a key factor that expands lysosomal functionality during young adulthood. Moreover, our findings suggest that the basal level of HLH-30 activity contributes to determining the rate of age-associated lysosomal dysfunction and subsequent decline in dendrite maintenance. Considering this together with the demonstrated therapeutic potential of TFEB activation in neurodegenerative disease models provides a strong foundation for future therapeutic strategies to stave off neuron aging and dysfunction [85,86,89].

## Materials and methods

### *C. elegans* strains and maintenance

For maintenance, *C. elegans* strains were grown on nematode growth medium (NGM) seeded with *E. coli* OP50 at room temperature (22 °C) [105]. Aged worms are defined by the number of days after the L4 stage, "Day 0" of adulthood. All experiments were performed at room temperature (22 °C), except for those in Figs 1E–1H, 2I, S2A, S4, and S5, which were performed at 20 °C. Wild-type N2 and JIN1375 *hlh-30(tm1978) IV* strains were obtained from the Caenorhabditis Genetics Center (CGC), which is funded by NIH Office of Research Infrastructure Programs (P40 OD010440). Strains used in this study are listed in S1 Table.

### Cloning and strain generation

Cloning was carried out in the pSM vector, a derivative of pPD49.26, or pPD117.01 (Addgene, Watertown, MA, USA) using standard restriction enzyme cloning technique (NEB, Ipswich, MA, USA). *Podr-1::gfp*, *Podr-1::rfp*, *Pdes-2::bfp*, and *Punc-122::rfp* vectors are kindly provided by the Kang Shen lab, Stanford University. *nuc-1, hlh-30,* and *rab-7* cDNA were amplified from a cDNA prepared from total RNA isolated from N2 worms. Sequence of the cloning and genotyping primers

are listed in S2 Table. Transgenes expressed from extrachromosomal arrays or integrated arrays were generated by microinjection of the constructs [106]. Genome editing was carried out by CRISPR-Cas9 [107]. The *hlh-30 syb* allele was generated at SunyBiotech (Fuzhou, Fujian, China) by CRISPR-Cas9.

**Confocal imaging and fluorescence microscopy**

Worms were transferred to a slide with a 3% agarose pad made by melting agarose in M9 buffer. Worms were immobilized in 10 mM sodium azide in M9 buffer for experiments to examine HLH-30::GFP localization and 10 mM levamisole in M9 buffer for all other experiments. Images were captured using a Nikon eclipse Ti2 microscope paired with a CSU W1 SoRa confocal scanner unit and a YOKOGAWA spinning disc unit with a Plan Apo VC 60xA/1.20 WI objective. Data for Figs 2D–2G, S2D, and S2E were collected using the 4× magnifier for Super-Resolution by Optical Re-Assignment. Image settings (laser power and exposure time) were identical for all genotypes and treatment groups across experiments. Images were taken with 0.5 μm step size (most images) or 0.4 μm step size (images with SNG-1::ARGO) in z-stack for 10–25 μm range. Images were analyzed using ImageJ software. Fluorescence intensities and sizes of vesicles were quantified from images using "3D object counter" (Figs 1B–1D, 1F, 2F–2H, 3F, and S3C). Mean fluorescence intensities were used for all figures and calculations except for *rab-7* transcription level (Fig 3F), which used maximum fluorescence intensities. Number of vesicles were quantified from images using "3D object counter" (Fig 2E), by hand (Figs 2B, 2C, and S2C) or using "find maxima" (S4C Fig). HLH-30 localizations were quantified from images by eye. PVD dendrite morphology were quantified by eye using the Nikon eclipse compound microscope.

For experiments using SNG-1::ARGO, maximum projection images were generated from z-stack of the whole presynaptic region of each animal. Presynapses were identified by generating a mask from the RFP image with default threshold settings followed by watershed. This mask was used to calculate the mean fluorescence intensity of each synapse in both the RFP and GFP images. The GFP/RFP was calculated for each presynapse detected, then averaged to generate one presynaptic GFP/RFP value for each animal.

Images of HLH-30 localization were taken in the green channel (Image setting: 100 ms exposure time, 60% laser power). The look-up table (LUT) of green channel was adjusted to 30−800 in ImageJ before quantifying. Categories of HLH-30 localization were performed as follow: neurons with HLH-30::GFP enriched in the cytoplasm and has a clear boundary between nucleus and cytoplasm were counted as enriched in the cytoplasm (enrich in C); neurons with HLH-30::GFP intensity in the cytoplasm that is higher than in the nucleus, but no clear boundary between nucleus and cytoplasm were counted as cytoplasm higher than nucleus (N < C); neurons with no difference in HLH-30::GFP intensity between nucleus and cytoplasm were counted as nucleus equal to cytoplasm (N = C).

Quantification of dendrite degeneration and sprouting was performed blind to genotype/condition.

To quantify dendrite degeneration, one dendrite from each worm was assigned to a category as follows: worms with dendrite that shows several swellings and the dendrite between swellings are relatively thinner than normal dendrites were counted as "slight dendrite degeneration;" worms with dendrite that have swellings that are disconnected from the main part of the dendrite were counted as "medium dendrite degeneration;" worms with dendrite that have more than five branches that are disconnected from the main part of the dendrite were counted as "severe dendrite generation;" worms without any feature mentioned above were counted as "no dendrite generation." Worms were always categorized by the most severe feature found in the whole dendrite.

To quantify dendrite sprouting, one dendrite from each worm was assigned to a category as follows: worms with disorganized dendrite outgrowth only at quaternary dendrites were counted as "slightly sprouting;" worms with disorganized dendrite outgrowth not only at quaternary dendrites, but restricted to one or two small areas were counted as "medium sprouting;" worms with disorganized dendrite outgrowth not only at quaternary dendrites among the whole dendrite were counted as "severe sprouting;" worms without any feature mentioned above were counted as "no sprouting."

## Concanamycin A treatment

NGM plates were supplemented to a final concentration of 0.03 µM concanamycin A (Chemcruz) from a 0.5 mM stock solution. Worms were moved onto seeded concanamycin A plates at the L4 stage/Day 0 of adulthood and were counted or imaged at Day 6 of adulthood. Of note, in performing experiments with concanamycin A-treated animals, we did not observe any detectable change in animal behavior caused by the drug. If the concanamycin A were having a strong impact on the neurons, we would expect it to impede the loading of neurotransmitter into synaptic vesicles, thereby causing aberrant, or uncoordinated movement. This is evidence that the amount of concanamycin A that the neurons experience is quite low.

For the concanamycin A injection, 25 µM concanamycin A in 5% DMSO, or 5% DMSO "mock drug," was injected into the pseudocoelom of *C. elegans*, together with 100 g/L orange G as a color tracer. Worms were allowed to recover for 3 hours before they were imaged as described above.

## Auxin treatment

NGM plates were supplemented to a final concentration of 1 mM K-NAA (PhytoTech Labs) from a 250 mM stock solution. Worms were synchronized by egg laying 15 gravid hermaphrodites for 2 h on seeded Auxin plates, and PVD dendrite morphology were counted at Day 9 unless otherwise noted.

## Heat shock treatment

For measuring HLH-30 localization under heat stress, worms growing at room temperature were shifted to 35 °C for 3 h at Day 3 of adulthood and were imaged immediately afterward (Fig 3A and 3B) [21]. For protein turnover experiments using SNG-1::ARGO, worms growing at 20 °C were shifted to 34 °C for 1 h at Day 2 of adulthood, returned to 20 °C, and then imaged 0–5 days afterward (S4 and S5 Figs) [56].

## Starvation treatment

Well-fed worms were moved to unseeded NGM plates at Day 2 of adulthood and were imaged at Day 3 of adulthood (Fig 3A and 3B).

## RNAseq data analysis

RNAseq data were obtained from NCBI's Gene Expression Omnibus. Sample "SRR19895471," "SRR19895472," "SRR19895473," "SRR19895474," "SRR19895475," "SRR19895476," "SRR19895477," "SRR19895478," and "SRR19895479" from project "PRJNA853940" were downloaded from NCBI [62]. Fastq reads were first trimmed and filtered for quality using TrimGalore (0.6.10). Reference genome were build using hisat2-build based on *C. elegans* Bristol N2 DNA reference. Each sample was aligned to the reference using hisat2 (2.1.0) and the results were organized using samtools sort (1.10). The raw gene hits were quantified using featureCounts (1.6.4). Raw gene hit count tables were normalized to total number of transcripts.

## Statistical analysis

Statistical analyses were performed in either Microsoft Excel, GraphPad Prism 10, or Rstudio. For quantitative data, single parametric pairwise comparisons were analyzed using unpaired two-tailed Student *t* test, and non-parametric pairwise comparisons were analyzed using Kolmogorov–Smirnov test. Multiple parametric comparisons were analyzed using one-way analysis of variance (ANOVA) test with Tukey post-test or two-way ANOVA with Sidak post-test as indicated in the figure captions. Data for Figs 1F, 2E–2G, S4E, and S4H were analyzed in R studio using a linear mixed-effects model as indicated in figure captions. For qualitative data, Fisher's exact test was performed between each individual comparison,

and Bonferroni or Holm correction were used to determine the significance threshold for the $P$ value. To compare SNG-1::ARGO half-lives between WT and *hlh-30(tm1978)*, Prism 10 was used to generate one-phase exponential decay curves, constraining the plateau to the experimentally determined value based on background GFP fluorescence (which comes from autofluorescence from the worm cuticle). The curves were fitted using least squares regression with no weighting, and the comparison between WT versus *hlh-30(tm1978)* was performed using Extra sum-of-squares $F$ test.

## Bootstrapping

Two hundred groups of 6 genes per group were randomly chosen from the RNAseq raw gene hits list. Raw gene hit count tables were normalized to total number of transcripts.

## Supporting information

**S1 Fig. Schematics showing the design of transgenes and alleles used. (A)** Lysosome acidity reporter, composed of three co-injected plasmids. **(B)** Lysosome degradative capacity reporter. The *nuc-1::rfp* and *bfp* morphology marker were co-injected, generating the transgene *carEx4*. The *Pnhr-81>Flippase* expresses in seam cells, including the PVD neuron's grandmother, and was expressed from *wyIs836* [50]. The *snt-1(ox698)* allele is FLP-on SNT-1::GFP [51]. **(C)** Design for the SNG-1::ARGO-tag [54]. **(D)** Design for fluorescent reporter of endogenous HLH-30 localization. **(E)** Design for *rab-7* transcriptional reporter, which was inserted into the *rab-7* endogenous genomic locus directly after the start codon. **(F)** Design of the *hlh-30::AID* allele and tissue-specific TIR1 expression transgenes for the Auxin-inducible degradation (AID) experiments.
(PDF)

**S2 Fig. Assessment of PVD neuron lysosomal compartments and animal life span in the *hlh-30* mutant. (A)** Example images of the NUC-1::RFP/GFP lysosomal acidity reporter in Day 9 adults. White dashed lines indicate neuron cell body and blue dashed lines outline lysosomal compartments. **(B)** Lifespan analysis of WT vs. the *hlh-30* mutant. \*$P < 0.05$, Kaplan–Meier. $n = 99$ (WT) and 98 (*hlh-30* mutant) animals spread across four plates. **(C)** The *hlh-30* mutant has fewer endogenously tagged GFP::RAB-7 endosomes in the PVD neuron cell body at Day 3 of adulthood compared to WT and shows no difference at Day 0. \*\*\*\*$P < 0.0001$, \*\*\*$P < 0.001$, ns: not significant, Two-way ANOVA with Sidak post-test. **(D)** Example images of lysosomal compartments in PVD neuron cell body in WT across adulthood. White dashed lines indicate the edge of the cell body. Scale bar = 10 μm. **(E)** Comparison of the NUC-1::RFP fluorescence from a Day 3 WT adult with standard confocal microscopy versus super resolution. Scale bar = 10 μm. **(F)** PVD neuron cell body size, quantified using *Pdes2>bfp,* is similar between Day 1 and Day 3 adults but significantly increases by Day 6 of adulthood. \*\*\*\*$P < 0.0001$, \*\*$P < 0.001$, \*$P < 0.05$, ns: not significant. One way ANOVA with Tukey post-test. The data underlying the graphs shown in the figure can be found in S1 Data.
(PDF)

**S3 Fig. Assessment of HLH-30 depletion using auxin inducible degradation. (A)** Example images showing the effect of tissue-specific TIR1 alleles + auxin on fluorescence of HLH-30::GFP11 + pan-somatically expressed GFP1-10. **(B, C)** Example images (B) and quantification (C) of control vs. auxin-treated animals carrying the pan-neuronal TIR1, the endogenously-tagged HLH-30::GFP11::AID allele, and a PVD>GFP1-10 transgene, which show that the pan-neuronal TIR1 strain depletes HLH-30 from the PVD neuron. Scale bar = 50 μm (A) or 10 μm (B). The data underlying the graphs shown in the figure can be found in S1 Data.
(PDF)

**S4 Fig. Synaptic protein Synaptogyrin/SNG-1 gets sorted into acidic compartment for degradation in the DA9 neuron. (A)** Schematic representation of the ARGO method for quantifying SNG-1 turnover using ratiometric fluorescence

imaging. In DA9 neurons, SNG-1 are tagged with both GFP and RFP before activation. After activation, newly synthesized SNG-1 proteins are tagged with only RFP. SNG-1 protein half-life can be measured by the decrease of relative GFP to RFP ratio over time. **(B–D)** DA9 presynapses shows no apparent difference in number or organization between WT and the *hlh-30* mutant (C), but the SNG-1::ARGO RFP fluorescence intensity is slightly decreased in the *hlh-30* mutant compared to WT (D). Scale bar = 10 μm. In (D), each data point shows the average of the maximum RFP intensity of all SNG-1::ARGO across all the presynapses in each worm. ***$P < 0.001$, **$P < 0.01$, ns: not significant, Two-way ANOVA with Tukey post-test. **(E, F)** Steady-state GFP/RFP ratio of SNG-1::ARGO in the cell body (E) and presynapses (F). The lower steady-state SNG-1::ARGO GFP/RFP ratio within vesicles in the cell body compared to at the synapses shows that the cell body vesicles are acidic lysosomal compartments. (E) The cell body SNG-1::ARGO GFP/RFP ratio is increased in the *hlh-30* mutant compared to WT at adult Day 4 and 7. Each data point shows the GFP/RFP ratio of SNG-1::ARGO-labeled endosome in the cell body. ****$P < 0.0001$, *$P < 0.05$, ns: not significant, Two-way ANOVA with Tukey post-test. (F) Steady-state SNG-1::ARGO GFP/RFP ratio at the synapses is decreased in the *hlh-30* mutant compared to WT at Day 2 and 4. Each data point is the average GFP/RFP from all presynapses within a single neuron, within a single animal. Note that the steady-state GFP/RFP ratio at the synapses is near two rather than one because GFP is brighter than RFP. ***$P < 0.001$, *$P < 0.05$, ns: not significant, linear mixed-effects model with Tukey post-test. Comparisons were performed on log-transformed data to meet model assumptions. (G-H) Day 3 worms injected with concanamycin A and imaged 3 hours post-injection showed an increased SNG-1::ARGO GFP/RFP ratio in the DA9 cell body (G) and an increased SNG-1::ARGO GFP/RFP ratio at the DA9 presynapses (H) compared to worms injected with mock drug. For (G), each data point shows the GFP/RFP ratio of SNG-1::ARGO-labeled endosome in the cell body. ***$P < 0.001$, two-tailed Student *t* test. For (H), each data point is the average GFP/RFP from all presynapses within a single neuron, within a single animal. ***$P < 0.001$, linear mixed-effects model; comparisons were performed on log-transformed data to meet model assumptions. The data underlying the graphs shown in the figure can be found in S1 Data. (PDF)

**S5 Fig. Loss of *hlh-30* delays the turnover of a characteristic cargo for lysosomal degradation, Synaptogyrin/SNG-1, in the DA9 neuron. (A)** Representative image of SNG-1::ARGO GFP, RFP channel in the DA9 cell body shows increased GFP fluorescence in aged *hlh-30* mutant. Scale bar = 10 μm. **(B)** Example images of presynaptic SNG-1::ARGO GFP, RFP, and the GFP/RFP ratio imaged at the indicated ages. For the Day 7 post-activation animal, activation occurred on Day 2 of adulthood. In the GFP/RFP ratio images, the color scale ranges from dark blue (low GFP/RFP) to pale yellow (high GFP/RFP). Scale bar = 10 μm. **(C)** Line scan images of the RFP channel in DA9 neuron synapses labeled by SNG-1::ARGO shows no apparent difference in synapse organization in the *hlh-30(tm1978)* mutant compared to WT. In the GFP/RFP ratio images, the color scale ranges from dark blue (low GFP/RFP) to pale yellow (high GFP/RFP). Scale bar = 10 μm. **(D, E)** The *hlh-30* mutant shows decreased SNG-1::ARGO puncta size (D) and fewer puncta (E) in the neuron cell body by Day 7 compared to WT. **$P < 0.01$, *$P < 0.05$, two-tailed Student *t* test. **(F, G)** Quantification of SNG-1::ARGO GFP/RFP ratio at the synapse after the heat shock "pulse" in WT (F) and the *hlh-30(tm1978)* mutant (G). Each data point shows the average GFP/RFP ratio from all presynapses within a single neuron. These data were used to generate the decay curves for Fig 2I. The data underlying the graphs shown in the figure can be found in S1 Data. (PDF)

**S6 Fig. mRNA abundance of HLH-30 target genes decreases with aging.** (A–E) In addition to *rab-7* (Fig 3G), three out of five more HLH-30-regulated genes show reduced mRNA level along with aging. **$P < 0.01$, *$P < 0.05$, ns: not significant, One-way ANOVA with Tukey post-test [60]. **(F)** Bootstrapping analysis showing numbers of genes in randomly selected groups of 6 genes that have reduced mRNA level with aging. The data underlying the graphs shown in the figure can be found in S1 Data. (PDF)

**S7 Fig. Model for how HLH-30 promotes dendrite maintenance.** In young adult animals, basal HLH-30 level increases to expand endo-lysosomal degradative capacity, maintaining dendrite integrity. In the *hlh-30* mutant or aged adult worms, reduced HLH-30 activity leads to inadequate lysosomal functions, causing aberrant dendrite morphology phenotypes in PVD neurons. Neuronal, muscular and hypodermal HLH-30 activity is required to protect the PVD dendrite from degeneration, whereas only neuronal HLH-30 helps prevent dendrite sprouting defect.
(PDF)

**S1 Table. *Caenorhabditis elegans* strains used in this paper.**
(DOCX)

**S2 Table. Primers used in this paper.**
(DOCX)

**S1 Data. Individual numerical values for figures.**
(XLSX)

## Acknowledgments

Some *C. elegans* strains were provided by the Caenorhabditis Genetics Center (CGC), the National BioResource Project, the Kang Shen lab, Stanford University, and the Erik Jorgensen lab, University of Utah. We thank Sophie Baumberger for generating strains, Adam Darlington for data analysis, Sherlyn Wijaya, Manuel Alvarez, Namita Khajanchi, Jiarong Gao, and Jingting Liang for feedback on the manuscript, and members of the Richardson lab for discussions. We thank Michael O. Harding from the Statistical Consulting group at the University of Wisconsin—Madison for assistance in the experimental design and data analysis for Figs 1, 2, and S4 Fig.

## Author contributions

**Conceptualization:** Claire E. Richardson.

**Data curation:** Ruiling Zhong.

**Formal analysis:** Ruiling Zhong, Claire E. Richardson.

**Funding acquisition:** Claire E. Richardson.

**Investigation:** Ruiling Zhong, Claire E. Richardson.

**Methodology:** Ruiling Zhong, Claire E. Richardson.

**Project administration:** Ruiling Zhong, Claire E. Richardson.

**Resources:** Claire E. Richardson.

**Supervision:** Claire E. Richardson.

**Validation:** Ruiling Zhong.

**Visualization:** Ruiling Zhong.

**Writing – original draft:** Ruiling Zhong, Claire E. Richardson.

**Writing – review & editing:** Ruiling Zhong, Claire E. Richardson.

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
