## [Editor Report · Decision Letter 0]

25 Nov 2024

Dear Dr Richardson,

Thank you for submitting your manuscript entitled "TFEB/HLH-30-mediated expansion of neuronal lysosomal capacity in early adulthood protects dendrite maintenance during aging in Caenorhabditis elegans" for consideration as a Short Report by PLOS Biology.

Your manuscript has now been evaluated by the PLOS Biology editorial staff as well as by an academic editor with relevant expertise and I am writing to let you know that we would like to send your submission out for external peer review.

Once your full submission is complete, your paper will undergo a series of checks in preparation for peer review. After your manuscript has passed the checks it will be sent out for review. To provide the metadata for your submission, please Login to Editorial Manager (https://www.editorialmanager.com/pbiology) within two working days, i.e. by Nov 28 2024 11:59PM.

Kind regards,

Ines

--

Ines Alvarez-Garcia, PhD

Senior Editor

PLOS Biology

---

## [Decision Letter · Decision Letter 1]

31 Jan 2025

Dear Dr Richardson,

Thank you for your patience while your manuscript entitled"TFEB/HLH-30-mediated expansion of neuronal lysosomal capacity in early adulthood protects dendrite maintenance during aging in Caenorhabditis elegans" was peer-reviewed at PLOS Biology. Please accept my apologies for the delay in sending you our decision. The manuscrpt has now been evaluated by the PLOS Biology editors, an Academic Editor with relevant expertise, and by three independent reviewers.

The reviews are attached below. As you will see, the reviewers find the conclusions interesting, but they also raise several issues that would need to be addressed before we can consider the manuscript further for publication. Reviewer 1 thinks you should include a comment on the different worm ages used for the different analyses, which vary significantly, and asks for quantification of several parameters, higher resolution figures to distinguish better the individual lysosomes, and missing controls. This reviewer also thinks you should refine the model to summarise better the findings and demonstrate that PVD-specific rescuing of HLH-30 attenuates the age-related lysosomal dysfunctions to fully support the model. Reviewer 2 asks you to assess lysosomal pH reduction with age and also the number of lysosomes and PVD cells size. Reviewer 3 only has minor suggestions.

In light of the reviews, we would like to invite you to revise the work to thoroughly address the reviewers' reports. Given the extent of revision needed, we cannot make a decision about publication until we have seen the revised manuscript and your response to the reviewers' comments. Your revised manuscript is likely to be sent for further evaluation by all or a subset of the reviewers.

**IMPORTANT - SUBMITTING YOUR REVISION**

3. Resubmission Checklist

a) *PLOS Data Policy*

b) *Published Peer Review*

Sincerely,

Ines

--

Ines Alvarez-Garcia, PhD

Senior Editor

PLOS Biology

Reviewers' comments

Rev. 1:

In this manuscript, the authors report age-related lysosomal dysfunction in C. elegans neurons, and discovered that the worm TFEB homolog, HLH-30, as a contributor to this change. They first generated fluorescent transgenic reporters to measure lysosomal pH-dependent quenching and cargo degradation in PVD neurons. Using these reporters, the authors found that lysosomal degradation decreases while lysosomal size increases with age. Next, they examined the involvement of HLH-30 and found that its loss accelerates lysosomal dysfunction during aging. Furthermore, the authors showed that HLH-30 nuclear localization is not affected by aging, but its transcriptional level decreases together with the transcriptional decrease of lysosomal genes. Lastly, they detected interesting age-related morphological changes in PVD dendrites, including increased degeneration and sprouting, and identified the potential role of HLH-30 and lysosomal dysfunction in these processes. Overall, the manuscript is well-written and presents interesting findings. However, the rigor of the results needs to be be strengthened, and several major concerns require further experimental evidence.

Major points:

1. The choice of worm ages for different analyses varies considerably across the manuscript, which makes it difficult to fully assess the temporal patterns of age-related changes in WT and mutant worms.

a. Morrimoto's prior work showed that neuronal dysfunction occurs as early as day 2 of adulthood (Ben-Zvi et al 2009). The authors should assess lysosomal activity starting at day 1 and include additional time points between day 6 and day 9 (in Figs. 1B, 1C, 1E, 2E). In addition, the authors should discuss whether the observed lysosomal dysfunction is a consequence of neuronal dysfunction, or a driver of it.

b. The rationale for choosing day 2 and day 7 in Figs. 2H, 2I is unclear. Data from additional time points should be provided.

c. Given that the lysosomal size and cargo accumulation difference become evident only between day 3 and day 9, the authors should include day 9 comparison in Figs. 2F, 2G beyond day 6. It is particularly important as no significant difference is observed in day 6 but trends appear.

d. It is shown that concanamycin A induces lysosomal dysfunction at day 6. Does this effect only occur at this later time point? Given that this is a crucial validation for the reporters, the authors should measure the effect of concanamycin A at other ages.

2. The authors should avoid reusing WT data for comparison. For comparing WT and the hlh-30(o) mutant, data from experiments conducted together should be shown, rather than reusing WT data from Fig 1.

3. In images shown in Fig. 1D, there are both normal and abnormal lysosomes in the same cell body. In Fig. 1E, day 9 samples seem to contain two populations. So, it might be informative to calculate the percentage of abnormal lysosomes per cell body at different ages.

4. Lysosomes larger than 10um appear unusually large and likely represent clusters as mentioned by the authors. This may indicate an age-related increase in lysosomal fusion/clustering. The authors should consider using higher-resolution microscopy to distinguish individual lysosomes more clearly.

5. When showing PVD rescuing effects, the authors should include a critical control without the hlh-30 mutant in the background. In addition, the authors should provide the statistical result for the comparison between day 0 and day 3 in the hlh-30(0) mutant. This comparison is important to determine whether there is still an age-related increase even without hlh-30. Furthermore, the authors should provide PVD rescuing results for SNT-1::GFP and NUC-1::1GFP/RFP ratio. These results are essential to determine whether PVD hlh-30 contributes to the observed lysosomal dysfunction.

6. In Fig. 3F, the authors should conduct age-related comparisons in the hlh-30(o) mutant, especially between day 3 and day 6. It seems that the age-related decrease also occurred without hlh-30. If so, it would suggest other factors contribute to the age-related decrease, while hlh-30 only mediates general decreases.

7. The transcriptional decrease of hlh-30 begins at day 3 (Fig. 3H), which is consistent with decreases in lmp-2, cpr-5, and syx -17 (Fig. S4B, D, E). However, the decrease of rab-7 transcription only started at day 6 (Fig. 3G). Together with point #6, it is likely that other factors may contribute to the age-related transcriptional decrease of rab-7.

8. It is clear from the data shown in Figure 4, dendrite degeneration and sprouting are regulated by different mechanisms: 1. PVD specific rescue of hlh-30 only rescued degeneration but not sprouting. 2. Neuronal, hypodermal, and muscular loss of HLH-30 all resulted in dendrite degeneration, while only neuronal loss of HLH-30 caused dendrite sprouting. 3. PVD specific overexpression of hlh-30 attenuated age-related dendrite sprouting but not its degeneration. Therefore, the authors should separate these two processes and refine the model to summarize their findings.

9. Without demonstrating that PVD-specific rescuing of HLH-30 attenuates the age-related lysosomal dysfunctions (see point #5), the proposed model cannot be fully supported. Especially, the dendrite degeneration does not appear directly linked to HLH-30 function in PVD, regardless of its role in regulating age-related lysosomal dysfunction.

Minor points:

1. "Consistent with the framework that endolysosomal protein degradation occurs in the neuron cell body, we observed endosomes with a lower SNG-1::ARGO GFP/RFP fluorescence intensity ratio in the cell body compared to the presynapses in WT animals." Please revise this sentence as the comparison was not conducted between cell body and presynapses, but between WT and hlh-30(0) mutant.

2. "Furthermore, in contrast to WT animals, we detected no difference in rab-7 transcription level between Day 0, Day 3, and Day 6 adults in hlh-30(o) mutants (Fig 3F)." This result was not shown.

3. There is no Figure S4G (line 368).

Rev. 2:

In this manuscript by Zhong and Richardson, the authors investigate the role of HLH-30, the C. elegans ortholog of mammalian transcription factor EB (TFEB), in maintaining neuronal health through the regulation of lysosomal function. While the study presents intriguing findings, including age-related changes in lysosomal compartment size and the role of hlh-30 in maintaining neuronal morphology, the conclusions are weakened by several critical issues. These include incomplete validation of experimental tools, missing control experiments, and claims about HLH-30-regulated gene expression trends with age that conflict with publicly available single-cell RNA-seq data. Furthermore, the description of HLH-30 localization and function under non-stress conditions requires revision for accuracy. Addressing these points would greatly enhance the manuscript's impact and its contribution to understanding HLH-30's role in lysosomal function and neuronal health.

Major comments:

Lysosome function declines during aging in C. elegans

- The ConcA validation experiment is important and can be used to assess reduced pH with age. Assessing the NUC-1::GFP/RFP ratio +/- ConcA in young animals (day 3 or 6) should show a larger increase after ConcA treatment compared to day 9 or 11, where the authors argue lysosomal pH has already increased. Additionally, ConcA treatment should also be applied to other reporters, including the SNG-1::GFP reporter and the RAB-7::mCherry reporter.

- The authors should also assess lysosome number (as in Fig 2D) +/- ConcA treatment with age

- The authors should also include the RAB-7::mCherry reporter +/- ConcA treatment with age as shown in Fig. 2A-B.

- Does PVD cell size change with age? Could age-related cell size changes correlate with the observed increase in lysosomal volume? PVD cell size should be measured using existing BFP images.

- Reassess the exposure levels for NUC-1::RFP in Fig. 1A and D and RAB-7::mCherry in Fig. 2A to ensure they are not overexposed. Include an outline of the cell body for all PVD neurons in Figs. 1 and 2 for clarity, as dendrites (?) are visible in Fig. 2, but not in Figure 1. Since this figure describes age-related changes in lysosomal compartments, include images of PVD neurons from aged animals (day 9 or 11) in Fig. 1A, similar to Fig. 1D.

HLH-30 is required for adult neuronal lysosomal capacity and function

- Provide an explanation for the observed discrepancy in Fig. 2D, where lysosome numbers are reduced >3-fold in hlh-30 mutants, despite no significant difference in NUC-1::RFP volume on day 3 of adulthood.

- Condense the discussion in lines 238-248 about age-related changes in lysosomal compartment size for wild-type and hlh-30 mutants, as the differences are minor and not statistically significant.

- In Fig. 2E, hlh-30 mutants are missing from analysis on day 9 and day 11. Provide an explanation for this.

- Replace all instances of "hlh-30(o)" in the manuscript with the full allele name "hlh-30(tm1978)" or refer to it as "hlh-30 deletion." Update all figures to use "hlh-30" for clarity and adherence to standard notation.

- Validate the SNG-1::ARGO GFP/RFP tool using ConcA as a control experiment to confirm its reliability in detecting lysosomal changes. Include this validation in Fig. 1. Once validated, use the tool in Fig. 2 to characterize lysosomal dynamics in wild-type and hlh-30 mutants.

Set-point HLH-30 activity in neurons operates without nuclear enrichment and declines in aging

- Clarify whether "set-point" refers to basal activity.

- The authors do not accurately describe the subcellular localization of C. elegans HLH-30 under non-stress conditions. Previously published data indicate that neither the endogenously GFP-tagged HLH-30 (PMID: 33314217, PMID: 39059513), the commonly used HLH-30::GFP overexpression strain (PMID: 28355222, PMID: 27001890, PMID: 23925298, and others), nor the neuron-specific HLH-30::GFP driven by the rab-3p promoter (PMID: 36419219) exhibit nuclear exclusion under basal conditions. This needs to be corrected throughout the manuscript. Notably, the current study indicates that HLH-30, when specifically expressed in PVD neurons, is excluded from the nucleus under basal conditions. This suggests the involvement of additional regulatory mechanisms unique to neurons. The authors should explicitly distinguish their newly developed HLH-30 strain from previously published strains and revise the manuscript. Furthermore, the authors should consider the possibility that, since HLH-30 is excluded from the nucleus in PVD cells, it could have a non-transcriptional/cytoplasmic role in promoting lysosomal function and neuronal health (e.g., TFEB can play a non-transcriptional role in mitochondrial regulation; see PMID: 3826332).

- In lines 106-108 of the introduction, the authors state that "Additionally, HLH-30-regulated gene expression shows a declining trend with age, suggesting that reduced HLH-30 activity contributes to age-associated lysosomal dysfunction." This claim is revisited in lines 358-373 and is based on their analysis of previously published whole-body RNA-seq data, which shows decreased expression of hlh-30 and one of its target genes, rab-7. However, single-cell RNA-seq (PMID: 37531250) and single nuclei RNA-seq (PMID: 38816550) data generated from C. elegans throughout the adult lifespan are available, and the data from PMID: 37531250 specifically indicate that hlh-30 expression increases with age, including in neurons. These findings conflict with the conclusions made in this manuscript and suggest that whole-body RNA-seq data may not accurately represent gene expression dynamics in specific cell types. The authors must update Fig. 3 with an analysis of publicly available single-cell and single-nuclei RNA-seq data to assess the expression of hlh-30 and its target genes with age specifically in PVD neurons, especially since they make conclusions on cell-autonomous functions of HLH-30. They should then reconsider and, if necessary, revise their claims based on these results.

- Figure 3F: Why is there no data point for the hlh-30 mutant on day 9?

Loss of HLH-30 accelerates age-related decline in dendrite morphology maintenance

- The images of the dendrite morphology are beautiful and we would like to encourage the authors to include the images of Figure S2 in figure 4 to fully illustrate the age-related defects in dendrite morphology.

- The authors claim that HLH-30 functions cell-intrinsically to promote healthy neuronal morphology with age. However, the tissue-specific rescue experiments in Fig. 4D and E reveal that PVD- and pan-neuronal HLH-30 rescue only partially restore neuronal degeneration and sprouting phenotypes. In contrast, pan-somatic rescue restores these phenotypes to wild-type levels, indicating that cell-non-autonomous effects of HLH-30 likely contribute to neuronal health.

- The authors should address the fact that HLH-30 expression levels are unknown in the rescue strains shown in Fig. 4D and E. This should be noted as a caveat in the discussion.

- While the authors claim HLH-30 has a neuronal-specific, cell-intrinsic role in maintaining neuronal morphology with age, data from Figs. 4H and I suggest that HLH-30 knockdown in other tissues, such as the hypodermis and muscle, also worsens degeneration and sprouting phenotypes. This indicates that HLH-30 expression in non-neuronal tissues contributes to neuronal health. Experimental assessment and discussion of the tissue-specific nuances of HLH-30's role in maintaining neuronal morphology with age are lacking. The authors may consider removing the data from non-neuronal rescue strains to focus on the neuronal role of HLH-30 while exploring the cell-non-autonomous role of HLH-30 in future studies.

- The efficiency of the AID system needs validation for each tissue-specific strain. Images and quantification of fluorescence intensity of HLH-30::GFP (or similar) under the corresponding tissue-specific promoters should be included in the supplementary materials. Validation is only necessary for the pan-somatic and pan-neuronal AID strains if data from other tissue-specific strains are removed.

- Day 1 controls should be included in the graphs for Fig. 4B and C to provide a clearer baseline for age-related changes.

- In lines 178 and 456-460, the authors state: "The hlh-30(o) mutant is reported to have a wild-type lifespan at 20°C and a slightly shortened lifespan at 25°C [19,74]. Importantly, the shortened lifespan at 25°C cannot be rescued with neuron-specific HLH-30 expression [74]. Therefore, the rescue of the hlh-30(o) mutant's dendrite degeneration and sprouting by PVD-specific or pan-neuronal hlh-30 expression is not an indirect effect of lifespan differences between the strains." However, other studies (PMID: 32302543 and PMID: 28198373) have reported a decreased lifespan for hlh-30(tm1978) at 20°C. The authors should measure the lifespan of hlh-30(tm1978) mutants under their experimental conditions to strengthen their conclusions and hypotheses.

- The authors need to further explain the normal function and relevance of cup-5 and the nature of the cup-5(lf) mutation. Currently, the description is limited to the statement: "We also assessed the PVD dendrite morphology in cup-5(lf) mutant animals, which are known to have disrupted lysosomal functions" (lines 423-425). Additional context regarding the molecular role of cup-5 and its mutation's impact on lysosomal function would clarify its significance in the study.

Minor comments:

- Suggestion to use color for fluorescent microscopy images for visual clarity (Figs. 1A and D, 2A and C, 3A and E).

- Fig S1 is not useful without transgene names.

- Fig. 2A: labeling of image 3 (hlh-30(o)_PVD>HLH-30::GFP) is not reader-friendly. Something as follows would be better: hlh-30(-) + HLH-30::GFP rescue in PVD.

- In Fig. 3D, include a heading or other additional labeling to indicate what gene the figure is referencing. Font size could also be significantly reduced.

- Indicate day of adulthood of assayed worms within the figure for Figs. 4D-K.

- Figs. 4K and J are not referenced in order in the text (lines 413-414).

- Many graphs would benefit from more detailed y-axis labeling (similar to Fig. 2B) or headings to improve readability. E.g., Fig. 2H, I - GFP/RFP ratio of what?; Fig. 4B-K - percentage of worms with what?

- Authors sould explain why they chose to assay worms on specific days of adulthood in the manuscript text, especially since days do not match across figures (e.g., days 1 and 3 in Fig. 2G; days 3, 6, 9 and 11 in Fig. 1; days 2 and 7 in Figs. 2H and I…).

- Keep graph labeling font consistent. Do not mix bold and plain text (e.g., Fig. 2B x-axis).

- Please indicate what figures each strain was used for in the strain list (Table 1).

Rev. 3:

This report demonstrates the role of the autophagy-related transcription factor hlh-30 in protecting neurons against the stress of aging. The authors developed interesting tools to define lysosomal function during the aging of C. elegans and show convincingly the negative effect of the loss of hlh-30 on neuronal health. The quality of the images of some of the reporters could be improved, which should address some of the issues with organelle quantification. In a few parts, some of the imaging is simplified to data points whereas representative images would remain informative. Unlike an increasing number of publications nowadays, the authors have done a thorough job of citing proper literature. Kudos to the authors, this is a well-written manuscript and an interesting study. Therefore, I recommend only a few minor revisions.

Minor comments:

In Fig. 1A and 1D, the red signal looks quite saturated and may have rendered the quantification more difficult. Possibly, lower exposure on confocal microscopy would be more suitable for quantification of small clusters of lysosomes, as noted on line 157-158. This is a relatively common issue with autophagy reporter as well that can be resolved by lower exposure.

Over-exposure seems to be an issue for the colocalization picture of NUC-1::RFP with SNT-1::GFP.

In Fig. 2D-L, the corresponding representative images for each condition should be made available in supplemental to help readers appreciate the difference (or lack thereof).

A schematic of the ARGO method could help readers better understand the approach.

The model, Figure 4L (note on line 476) is missing.

---

## [Decision Letter · Decision Letter 2]

12 Jun 2025

Dear Dr Richardson,

Thank you for your patience while we considered your revised manuscript entitled "TFEB/HLH-30-mediated expansion of neuronal lysosomal capacity in early adulthood protects dendrite maintenance during aging in Caenorhabditis elegans" for consideration as a Short Report at PLOS Biology. Your revised study has now been evaluated by the PLOS Biology editors, the Academic Editor and the two original reviewers.

The reviews are attached below. You will see that while Reviewer 1 is now completely satisfied, Reviewer 2 raises two remainig points and thinks that you would need to perform further experiments to validate the use of ConcA as a tool and address the limitations raised. In addition, the reviewer mentions that while it is mentioned that single-nucleus RNA-seq data don’t support the claim of age-related declined in lysosomal gene expression, this discrepancy is not resolved.

In light of the reviews, we are pleased to offer you the opportunity to address the remaining points raised by Reviewer 2 in a revision that we anticipate should not take you very long. Please note that after discussing the points with the Academic Editor, we would like to see validation of the use of ConcA as a tool, but we are not concerned about the discrepancies of the age-related declined in lysosomal gene expression. We will then assess your revised manuscript and your response to the reviewers' comments with our Academic Editor aiming to avoid further rounds of peer-review, although we might need to consult with the reviewers, depending on the nature of the revisions.

We expect to receive your revised manuscript within 1 month. Please email us (plosbiology@plos.org) if you have any questions or concerns, or would like to request an extension. At this stage, your manuscript remains formally under active consideration at our journal; please notify us by email if you do not intend to submit a revision so that we withdraw the manuscript.

**IMPORTANT - SUBMITTING YOUR REVISION**

3. Resubmission Checklist

Sincerely,

Ines

--

Ines Alvarez-Garcia, PhD

Senior Editor

PLOS Biology

Reviewers' comments

Rev. 1:

The authors have fully addressed my comments.

Rev. 2:

The authors have put substantial effort into revising the experimental work, but the interpretive and narrative update is incomplete. New data were added but not sufficiently integrated into the overall story. Most notably, the use of Concanamycin A (ConcA) as a pharmacological tool is not rigorously validated. Despite relying on ConcA to support their claims about lysosomal dysfunction, the authors acknowledge multiple times that it produces limited or no effect in their assays and offer speculative explanations, such as poor cuticle permeability or neuron-specific resistance, without experimentally addressing these limitations. A proper validation would require a time-course analysis, multiple doses, positive controls (e.g., injected ConcA), or use of sensitized mutant backgrounds known to enhance drug uptake. Instead, the manuscript includes only superficial treatment of these issues, and yet still draws conclusions based on the assumption that ConcA is functioning as expected. This is especially problematic in light of the failed effects in key reporter assays such as SNG-1::ARGO. In addition, the authors briefly mention published single-nucleus RNA-seq data that do not support their claim of age-related declines in lysosomal gene expression but do not explore or visualize this discrepancy. This undermines the strength of their conclusions and misses an opportunity for rigorous comparison.

---

## [Editor Report · Decision Letter 3]

12 Aug 2025

Dear Dr Richardson,

Thank you for your patience while we considered your revised manuscript entitled "TFEB/HLH-30-mediated expansion of neuronal lysosomal capacity in early adulthood protects dendrite maintenance during aging in Caenorhabditis elegans" for publication as a Short Report at PLOS Biology. This revised version of your manuscript has been evaluated by the PLOS Biology editors and the Academic Editor.

Based on our Academic Editor's assessment of your revision, we are likely to accept this manuscript for publication, provided you satisfactorily address the data and other policy-related requests stated below my signature.

Please also note that Short Reports only have four figures, thus you have convert two of the main figures into supplementary figures.

In addition, we would like you to consider a suggestion to improve the title:

"Expansion of lysosomal capacity in early adult neurons driven by TFEB/HLH-30 protects dendrite maintenance during aging in Caenorhabditis elegans"

We expect to receive your revised manuscript within two weeks.

*Published Peer Review History*

*Press*

Sincerely,

Ines

--

Ines Alvarez-Garcia, PhD

Senior Editor

PLOS Biology

Fig. 1B, C, D, F-H; Fig. 2B, C, F-I; Fig. 3C-H, J-L; Fig. 4B, F, G, H; Fig 5B, C, D-M (should be supplementary); Fig. S2C, D, E; Fig. S3C; Fig. S4C, D and Fig. S5A-F

**Please also make publicly available at this stage the data you have deposited at the BioImage Archive and provide the accession number.

CODE POLICY

---

## [Editor Report · Decision Letter 4]

18 Sep 2025

Dear Dr Richardson,

Thank you for the submission of your revised Short Report entitled "Expansion of lysosomal capacity in early adult neurons driven by TFEB/HLH-30 protects dendrite maintenance during aging in Caenorhabditis elegans" for publication in PLOS Biology. On behalf of my colleagues and the Academic Editor, Maria Fernanda Ceriani, I am delighted to let you know that we can in principle accept your manuscript for publication, provided you address any remaining formatting and reporting issues. These will be detailed in an email you should receive within 2-3 business days from our colleagues in the journal operations team; no action is required from you until then. Please note that we will not be able to formally accept your manuscript and schedule it for publication until you have completed any requested changes.

PRESS

Sincerely, 

Ines

--

Ines Alvarez-Garcia, PhD

Senior Editor

PLOS Biology
